# ALIFE: Adaptive Logit Regularizer and Feature Replay for Incremental Semantic Segmentation

**Youngmin Oh**      **Donghyeon Baek**      **Bumsub Ham**[*]
School of Electrical and Electronic Engineering, Yonsei University
https://cvlab.yonsei.ac.kr/projects/ALIFE

## Abstract

We address the problem of incremental semantic segmentation (ISS) recognizing novel object/stuff categories continually without forgetting previous ones that have been learned. The catastrophic forgetting problem is particularly severe in ISS, since pixel-level ground-truth labels are available only for the novel categories at training time. To address the problem, regularization-based methods exploit probability calibration techniques to learn semantic information from unlabeled pixels. While such techniques are effective, there is still a lack of theoretical understanding of them. Replay-based methods propose to memorize a small set of images for previous categories. They achieve state-of-the-art performance at the cost of large memory footprint. We propose in this paper a novel ISS method, dubbed ALIFE, that provides a better compromise between accuracy and efficiency. To this end, we first show an in-depth analysis on the calibration techniques to better understand the effects on ISS. Based on this, we then introduce an adaptive logit regularizer (ALI) that enables our model to better learn new categories, while retaining knowledge for previous ones. We also present a feature replay scheme that memorizes features, instead of images directly, in order to reduce memory requirements significantly. Since a feature extractor is changed continually, memorized features should also be updated at every incremental stage. To handle this, we introduce category-specific rotation matrices updating the features for each category separately. We demonstrate the effectiveness of our approach with extensive experiments on standard ISS benchmarks, and show that our method achieves a better trade-off in terms of accuracy and efficiency.

## 1  Introduction

Humans are capable of learning new concepts continually, while preserving or even improving previously acquired knowledge. Artificial neural networks are, however, prone to forget the knowledge they have learned if being trained with samples for new object/scene categories alone. The reason for this problem, so-called catastrophic forgetting [11, 27], is that parameters of neural networks change abruptly to handle new categories without accessing training samples for previous categories. A straightforward way to alleviate the problem is to re-train a model with training examples for entire categories observed so far, which is however computationally demanding.

Incremental learning is an alternative approach to learning new categories continuously without re-training on the entire dataset. While many methods have been proposed for incremental classification [4, 21, 23, 25, 33, 36], a few attempts explore incremental semantic segmentation (ISS), where training images for new categories are partially labeled to reduce the cost for manual annotation. That is, pixels for new categories are labeled only, while remaining ones are marked as unknown. The unknown regions should be considered separately, since they could contain previous categories along with ones that would be seen in the future.

---

[*]Corresponding author.

36th Conference on Neural Information Processing Systems (NeurIPS 2022).

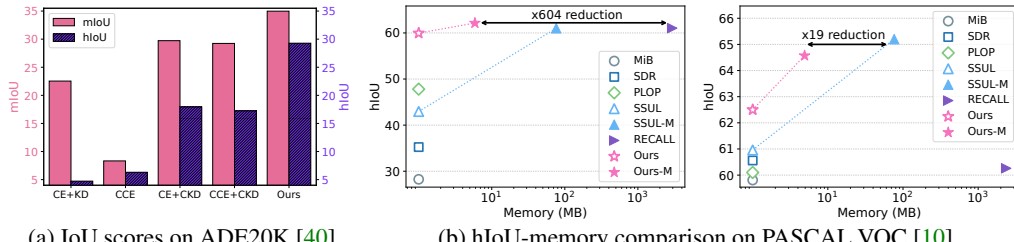

(a) IoU scores on ADE20K [40].   (b) hIoU-memory comparison on PASCAL VOC [10].

Figure 1: (a) Quantitative comparison of intersection-of-union (IoU) scores on ADE20K. Each model learns 50 novel categories after learning 100 categories. To be specific, there is a total of 5 incremental stages, and each model learns 10 new categories at every incremental stage. mIoU: an average IoU score for 150 categories. hIoU: a harmonic mean between two average IoU scores for previous (100) and novel (50) categories. (b) Quantitative comparison of hIoU and memory requirement on PASCAL VOC. (From left to right) Each method learns 1 and 5 novel categories after learning 20 and 16 categories, respectively. Ours-M memorizes 1K features for each previous category. For RECALL [26] and SSUL-M [3], we consider the size of memory required to store images only, and discard that for corresponding labels or saliency maps. Best viewed in color.

Current ISS methods can be categorized into regularization-based [2, 9, 30] and replay-based approaches [3, 26]. Regularization-based methods typically exploit knowledge distillation (KD) [15] to preserve the discriminative ability of ISS models for previous categories. In particular, the seminal work of [2] alleviates the semantic shift of unknown regions. Specifically, it introduces calibrated cross-entropy (CCE) and calibrated KD (CKD) terms to learn semantic information from unknown regions. Although CCE and CKD terms have shown the effectiveness on ISS methods [2, 9, 30], there is still a lack of theoretical understanding. Replay-based methods propose to use a replay buffer consisting of web-crawled or previously seen images. The replay buffer provides rich information storing the knowledge for previous categories, but it incurs large memory footprint, particularly for the task of semantic segmentation that typically adopts high-resolution images.

In this paper, we present a novel ISS method, dubbed ALIFE, that 1) alleviates catastrophic forgetting and 2) reduces memory requirements. For the first aspect, we analyze gradients of CCE and CKD terms for better understanding the effectiveness on catastrophic forgetting, and make the following observations: (1) For unknown regions, CCE reduces logit values of new categories, which is crucial for preventing overfitting to the new categories. However, it always raises logit values of all previous categories, without considering whether predictions for those regions are correct or not, which lessens the discriminative power for previous categories. (2) CKD makes it difficult to distinguish new categories from a background one. Motivated by these observations, we introduce an adaptive logit regularizer (ALI) that enables better learning new categories and alleviating catastrophic forgetting for previous ones (Fig. 1(a)). For the second aspect, we propose to exploit latent features for replaying, reducing memory footprint and avoiding data privacy issues (*e.g.*, in medical imaging [35]). In contrast to the replay buffer storing images directly, our approach to using the features updates them at every incremental stage, as a feature extractor is also updated continually at training time. Specifically, we exploit category-specific rotation matrices using the Cayley transform. Rotating latent features is computationally efficient, while maintaining the relations between the features. We demonstrate that exploiting ALI with memorizing features achieves a better trade-off in terms of accuracy and efficiency (Fig. 1(b)). Our main contributions are summarized as follows:

- We show an in-depth analysis of probability calibration methods widely used for ISS [2, 9, 30], and introduce ALI that enables our model to better learn new categories, while maintaining the knowledge for previous categories.

- We present a novel replay strategy using category-specific rotation matrices, which helps to alleviate catastrophic forgetting for previous categories with much less memory requirements than replaying raw images.

- Extensive experiments demonstrate the effectiveness of our approach to using ALI and replaying features with rotation matrices. We set a new state of the art on standard ISS benchmarks [10, 40].

## 2 Related work

Many incremental learning approaches have been introduced to preserve knowledge for previous categories, while recognizing new ones [28]. They can be categorized into task-incremental and class-incremental methods. Task-incremental methods [21, 23, 25] treat learning a new set of categories as a new task. With a perfect task identifier, they classify each task separately at test time. The perfect identifier is however often not available in practice [32]. On the contrary, class-incremental methods [4, 8, 33, 36] attempt to recognize all categories observed so far at once without the task identifier, being more practical. In the following, we describe class-incremental methods pertinent to ours.

**Image classification.**   Incremental methods typically adopt a KD technique [15] in order to encourage a current model to imitate softmax probabilities [1, 33] or intermediate representations [8, 18] obtained from a previous one, retaining the discriminability for previous categories. Another line of works focuses on preventing a classifier from overfitting to new categories, by correcting biased weights of a classifier with a post-processing [36, 39] or normalizing a classifier [18]. Similarly, our ALI is beneficial to alleviating the overfitting problem and maintaining the discriminative power for previous categories, but differs in that it adaptively regularizes logit values during training. In addition, we exploit latent features for replaying, while all the aforementioned methods rely on an image replay strategy that entails practical issues such as data privacy and large memory requirements. To avoid these problems, recent works employ a pseudo replay scheme that synthesizes images [20, 31, 34] or latent features [38] of previous categories. However, they often require a generator [12], which should also be trained incrementally. Instead of using raw images or synthesizing features, the work of [19] proposes to store latent features. It trains a feature adaptation network (FAN) mapping the latent features into a new feature space. The FAN might be sub-optimal, since the same network applies to all features without differentiating categories. Adopting individual FANs for each category might address the problem, but this requires high computational cost and large memory footprint. Differently, we propose to train category-specific rotation matrices. This is more efficient and accurate, as a rotation matrix is light-weight and features of each category are updated separately.

**Semantic segmentation.**   A relatively few methods address the task of ISS, which can further be classified into regularization-based and replay-based approaches. The first approaches [2, 9, 30] focus on preserving knowledge for previous categories without using an experience replay. For example, MiB [2] proposes to consider the semantic shift of a background category in ISS, and introduces CCE and CKD terms using probability calibration techniques. While the calibration techniques are intuitive and effective, there are still lack of theoretical explanations. We provide a gradient analysis on CCE and CKD, and point out that CCE and CKD could disturb preserving knowledge for previous categories and discriminating new categories from the background category, respectively. SDR [30] and PLOP [9] propose to further regularize a current model in a latent feature space. Specifically, SDR minimizes distances between features of the same category [6], while PLOP distills intermediate features using a multi-scale strip pooling technique [8, 17]. The second approaches [3, 26] rely on an image replay strategy. RECALL [26] exploits web-crawled images with pseudo labels at each incremental stage, while SSUL [3] maintains a small set of images for previous categories, together with corresponding ground-truth masks. Both methods show state-of-the-art results at the cost of large memory footprint. Our approach differs in two aspects: (1) We exploit latent features for replaying, which reduces the size of required memory significantly; (2) Our method updates a feature extractor continually, and thus it is more flexible than RECALL and SSUL freezing the extractor.

## 3 Method

In this section, we first describe the ISS task briefly (Sec. 3.1), and then present our approach consisting of three steps at every incremental stage. Specifically, we introduce ALI based on an in-depth analysis for CCE and CKD to train a ISS model (Sec. 3.2) and present a feature replay scheme using category-specific rotation matrices (Sec.3.3) to fine-tune a classifier using memorized features (Sec. 3.4). Note that the last two steps are optional. Please refer to the supplementary material for detailed derivations, more analysis, and a pseudo code of our approach.

### 3.1 Problem statement

Following the common practice [2, 3, 9, 26, 30], we consider a series of $T$ learning stages. Each stage $t$ has its own dataset, $D^t = \{(x_i, y_i)\}_{i=1}^{n^t}$ of size $n^t$, where $x$ and $y$ are an image and a

corresponding ground-truth mask, respectively. For incremental stages (*i.e.*, $t > 1$), we have two disjoint sets, $C_{\text{prev}}^t$ and $C_{\text{new}}^t$, for previously learned and novel categories, respectively. Note that ground-truth masks are labeled only for the categories of $C_{\text{new}}^t$. Formally, we denote by $y(\mathbf{p})$ a ground-truth label at position $\mathbf{p}$, and define labeled regions as follows:

$$\mathcal{R}_{\text{new}}^t = \bigcup_{c \in C_{\text{new}}^t} \mathcal{R}_c, \tag{1}$$

where $\mathcal{R}_c$ is a set of locations labeled as a category $c$, that is, $\mathcal{R}_c = \{\mathbf{p} \mid y(\mathbf{p}) = c\}$.

Our goal is to train a model that recognizes all categories observed so far, $C_{\text{all}}^t$, the union of the disjoint sets, *i.e.*, $C_{\text{all}}^t = C_{\text{prev}}^t \cup C_{\text{new}}^t$. In detail, our model consists of a feature extractor $\phi$ and a classifier $w$. The feature extractor takes an image and outputs a convolutional feature map for prediction, $\phi^t : x \mapsto f^t$, where we denote by $f^t(\mathbf{p}) \in \mathbb{R}^D$ a $D$-dimensional feature vector at position $\mathbf{p}$. A logit value for a category $c$ is then computed by the dot product between a feature and a classifier weight for the category $c$, $w_c^t \in \mathbb{R}^D$, as follows:

$$z_c^t(\mathbf{p}) = w_c^t \cdot f^t(\mathbf{p}), \tag{2}$$

where we omit bias terms of the classifier for brevity.

### 3.2 Step 1

In the first step, we initialize network weights of a model at the stage $t$ with those for the previous stage $t - 1$, and train a feature extractor and a classifier of the current model with a corresponding dataset $D^t$. A simple way to mitigate catastrophic forgetting for ISS is employing cross-entropy (CE) and KD [15] terms. To this end, we compute probabilities for a category $c$ at position $\mathbf{p}$ as follows:

$$p_c^t(\mathbf{p}) = \frac{e^{z_c^t(\mathbf{p})}}{\sum_{k \in C_{\text{all}}^t} e^{z_k^t(\mathbf{p})}}, \quad c \in C_{\text{all}}^t \quad \text{and} \quad q_c^t(\mathbf{p}) = \frac{e^{z_c^t(\mathbf{p})}}{\sum_{k \in C_{\text{prev}}^t} e^{z_k^t(\mathbf{p})}}, \quad c \in C_{\text{prev}}^t, \tag{3}$$

which are obtained by applying the softmax function to logit values across $C_{\text{all}}^t$ and $C_{\text{prev}}^t$, respectively. The CE and KD terms for ISS are then defined as follows:

$$L_{\text{CE}}(\mathbf{p}) = -\log p_{c^*}^t(\mathbf{p}), \quad \mathbf{p} \in \mathcal{R}_{\text{new}}^t \quad \text{and} \quad L_{\text{KD}}(\mathbf{p}) = \sum_{k \in C_{\text{prev}}^t} -p_k^{t-1}(\mathbf{p}) \log q_k^t(\mathbf{p}), \quad \forall \mathbf{p}, \tag{4}$$

where $c^* = y(\mathbf{p})$. The CE term is defined for the labeled regions $\mathcal{R}_{\text{new}}^t$ only, indicating that logit values of previous categories always decrease. Exploiting the CE term alone is thus prone to overfitting to new categories and leads to catastrophic forgetting. The KD term addresses this problem by transferring the discriminative power of a previous model, trained to classify previous categories, into the current one. Note that $C_{\text{prev}}^t = C_{\text{all}}^{t-1}$, and the KD term computes softmax probabilities across previous categories without considering new ones. Instead of exploiting CE and KD terms, the seminal work [2] proposes to use CCE and CKD, widely adopted in ISS [2, 9, 30]. In the following, we analyze gradients of CCE and CKD to better understand the influences on ISS, and introduce ALI to train our model.

**CCE.** The CCE loss [2] additionally computes a probability for unlabeled regions by summing the probabilities over all previous categories as follows:

$$L_{\text{CCE}}(\mathbf{p}) = \begin{cases} -\log p_{c^*}^t(\mathbf{p}), & \mathbf{p} \in \mathcal{R}_{\text{new}}^t \\ -\log p_{\text{cce}}^t(\mathbf{p}), & \mathbf{p} \notin \mathcal{R}_{\text{new}}^t \end{cases}, \tag{5}$$

where $p_{\text{cce}}^t(\mathbf{p}) = \sum_{k \in C_{\text{prev}}^t} p_k^t(\mathbf{p})$. This marginal difference

Table 1: Gradients of CCE w.r.t $z_c^t$. Note that the optimization process is carried out by gradient descent.

| Conditions | | Gradients |
|---|---|---|
| $\mathbf{p} \in \mathcal{R}_{\text{new}}^t$ | $c = y(\mathbf{p})$ | $p_c^t - 1$ |
| | $c \neq y(\mathbf{p})$ | $p_c^t$ |
| $\mathbf{p} \notin \mathcal{R}_{\text{new}}^t$ | $c \in C_{\text{new}}^t$ | $p_c^t$ |
| | $c \in C_{\text{prev}}^t$ | $p_c^t - q_c^t$ |

from the vanilla CE term brings a significant improvement on ISS. To analyze the reason behind the improvement, we summarize in Table 1 gradients of $L_{\text{CCE}}$ w.r.t $z_c^t$. The first two rows show that CCE updates logit values for the labeled regions in the same way as in $L_{\text{CE}}$. We can see from last two rows that CCE also provides gradients for the unlabeled regions. Specifically, the third row shows that CCE reduces a logit value for new categories by the corresponding probability $p_c^t$, if $\mathbf{p} \notin \mathcal{R}_{\text{new}}^t$. This is reasonable, since all features on those regions do

not at least belong to new categories. CCE thus helps to avoid the overfitting problem, alleviating catastrophic forgetting. Considering the fact that $q_c^t$ is always larger than $p_c^t$ (See the supplementary material), we can see from the last row that CCE raises logit values for all previous categories by $q_c^t - p_c^t$ in the unlabeled regions. This prevents the logit values of previous categories from continuing to decrease in the unlabeled regions, which is however effective only when predictions of the current model are correct. Otherwise, when predictions are incorrect, CCE rather raises logit values of the wrongly predicted categories, aggravating the incorrect predictions.

**CKD.** Assuming that categories of $C_{\text{new}}^t$ are likely to be labeled as a background one in the previous stage, the CKD loss [2] is defined as follows:

$$L_{\text{CKD}}(\mathbf{p}) = -p_{bg}^{t-1}(\mathbf{p}) \log p_{\text{ckd}}^t(\mathbf{p}) + \sum_{k \in C_{\text{prev}}^t \setminus \{bg\}} -p_k^{t-1}(\mathbf{p}) \log p_k^t(\mathbf{p}), \quad \forall \mathbf{p}, \tag{6}$$

where $p_{\text{ckd}}^t(\mathbf{p}) = \sum_{k \in \{bg\} \cup C_{\text{new}}^t} p_k^t(\mathbf{p})$. For previous categories except the background, this term enables transferring knowledge from $p_k^{t-1}$ to $p_k^t$ directly. Similarly, a vanilla KD term in Eq. (4) also encourages a current model to output probabilities similar to $p_k^{t-1}$ for $k \in C_{\text{prev}}^t$, but it distills the knowledge from $p_k^{t-1}$ into $q_k^t$ without considering logit values for new categories. To compare KD and CKD, we compute gradients w.r.t $z_c^t$ for $c \in C_{\text{prev}}^t \setminus \{bg\}$ as follows:

$$\frac{\partial L_{\text{KD}}(\mathbf{p})}{\partial z_c^t(\mathbf{p})} = q_c^t(\mathbf{p}) - p_c^{t-1}(\mathbf{p}) \quad \text{and} \quad \frac{\partial L_{\text{CKD}}(\mathbf{p})}{\partial z_c^t(\mathbf{p})} = p_c^t(\mathbf{p}) - p_c^{t-1}(\mathbf{p}). \tag{7}$$

Here we provide two cases to explain the reason why CKD better transfers the knowledge for previous categories except the background: (1) When $q_c^t$ and $p_c^{t-1}$ are equal, the gradient of KD w.r.t $z_c^t$ is zero, indicating that $z_c^t$ remains the same. However, note that $p_c^t$ is lower than $p_c^{t-1}$ in this case, since $p_c^t$ is always lower than $q_c^t$. This suggests that $z_c^t$ should rather increase in order that $p_c^t$ follows the target probability $p_c^{t-1}$. The gradient of CKD w.r.t $z_c^t$ has a negative value in the same case (*i.e.*, $q_c^t = p_c^{t-1}$), suggesting that $z_c^t$ increases by gradient descent. (2) When the current model already provides the same probability as the target one (*i.e.*, $p_c^t = p_c^{t-1}$), it is unnecessary to adjust $z_c^t$. The gradient of KD w.r.t $z_c^t$ has a positive value, as $q_c^t$ is larger than $p_c^t$ (or equivalently $p_c^{t-1}$ in this case). This indicates that $z_c^t$ decreases by gradient descent, and $p_c^t$ in turn becomes lower than $p_c^{t-1}$. On the other hand, CKD maintains $z_c^t$ the same, since its gradient w.r.t $z_c^t$ is zero. These examples describe the effectiveness of CKD. On the contrary, CKD also has negative effects on ISS. Let us suppose the gradients of CKD w.r.t $z_c^t$ for $c \in \{bg\} \cup C_{\text{new}}^t$ as follows:

$$\frac{\partial L_{\text{CKD}}(\mathbf{p})}{\partial z_c^t(\mathbf{p})} = \left( p_{\text{ckd}}^t(\mathbf{p}) - p_{bg}^{t-1}(\mathbf{p}) \right) \frac{p_c^t(\mathbf{p})}{p_{\text{ckd}}^t(\mathbf{p})}. \tag{8}$$

The problem of CKD lies in how it transfers the knowledge from $p_{bg}^{t-1}$ into $p_{bg}^t$. In particular, CKD makes $p_{\text{ckd}}^t$ to imitate $p_{bg}^{t-1}$, instead of directly distilling from $p_{bg}^{t-1}$ to $p_{bg}^t$. For example, when $p_{\text{ckd}}^t$ is lower than $p_{bg}^{t-1}$, the gradient of CKD in Eq. (8) has a negative value. Thus, logit values for background and new categories increase by gradient descent. This in turn raises probabilities for background and new categories all together, disturbing discriminating the new categories from the background at training time. In case of $p_{\text{ckd}}^t > p_{bg}^{t-1}$, CKD reduces the logit values for background and new categories, which is however problematic when $p_{bg}^t < p_{bg}^{t-1}$. Note that the logit value for the background category $z_{bg}^t$ should rather increase in this case.

**ALI.** We have shown that CCE alleviates catastrophic forgetting, while CKD better guides transferring the knowledge of a previous model than KD. To avoid the negative effects of CCE and CKD, we present in Table 2 a new form of gradients w.r.t $z_c^t$. The first row is identical to the third one of Table 1

Table 2: Gradients of ALI w.r.t $z_c^t$.

| Conditions | | Gradients |
|---|---|---|
| $\mathbf{p} \notin \mathcal{R}_{\text{new}}^t$ | $c \in C_{\text{new}}^t$ | $p_c^t$ |
| | $c \in C_{\text{prev}}^t$ | $p_c^t - p_c^{t-1}$ |

that alleviates catastrophic forgetting. The second row is similar to the gradients of CKD in Eq. (7) that better capture knowledge for previous categories. Note that CKD excludes the background category in Eq. (7), since it assumes that new categories of the current stage ($C_{\text{new}}^t$) are marked as the background one at the previous stage. Differently, ours enables computing gradients for all

previous categories including the background one (See the second row), since it does not require the assumption. Accordingly, this form of gradients incorporates the advantages of CCE and CKD, while discarding the negative effects. Integrating the gradients in Table 2 w.r.t $z_c^t$, we define ALI as follows:

$$L_{\text{ALI}}(\mathbf{p}) = \log\left(\sum_{k \in C_{\text{all}}^t} e^{z_k^t(\mathbf{p})}\right) - \sum_{k \in C_{\text{prev}}^t} p_k^{t-1}(\mathbf{p}) z_k^t(\mathbf{p}), \quad \mathbf{p} \notin \mathcal{R}_{\text{new}}^t. \tag{9}$$

The first term applies the log-sum-exp function to logit values across $C_{\text{all}}^t$, approximating the maximum logit value over $C_{\text{all}}^t$. The second term computes a weighted average of logit values for $C_{\text{prev}}^t$ with probabilities of a previous model. In this context, ALI can be viewed as minimizing the difference between the maximum logit value and the weighted average adaptively. That is, it reduces the maximum logit value, while raising the logit values for previous categories. Note that ALI does not require computing either $p_{\text{cce}}^t$ in CCE or $p_{\text{ckd}}^t$ in CKD [2].

**Training.** To train our model, we use CE and ALI terms for labeled and unlabeled regions, respectively. We also apply a vanilla KD term for labeled regions to further regularize our model. An overall objective for the first step is defined as follows:

$$L_{\text{S1}}(\mathbf{p}) = L_{\text{CE}}(\mathbf{p}) + \lambda_{\text{ALI}} L_{\text{ALI}}(\mathbf{p}) + \lambda_{\text{KD}} L_{\text{KD}}(\mathbf{p}) \mathbb{1}[\mathbf{p} \in \mathcal{R}_{\text{new}}^t], \tag{10}$$

where $\lambda_{\text{ALI}}$ and $\lambda_{\text{KD}}$ are balance parameters. We denote by $\mathbb{1}[\cdot]$ an indicator function whose output is 1 if the argument is true, and 0 otherwise.

### 3.3 Step 2

After training our model, we first extract features of new categories in order to replay them in subsequent stages. Then, we compensate a distribution shift of memorized features, which are extracted in the previous stage $t-1$, before using them to fine-tune a classifier in the third step. To be specific, we exploit category-specific rotation matrices to update memorized features of each category separately. In the following, we provide detailed descriptions for memorizing features and training matrices. Note that we freeze both a feature extractor $\phi^t$ and a classifier $w^t$ for the second step.

**Memorizing features.** We extract features for prediction (*i.e.*, $f^t$) and store them to replay in subsequent stages. Specifically, given input images containing one of new categories, a feature extractor first produces feature maps. For each image, features for each category are then averaged using a ground-truth mask. This process is repeated until the number of features for each category reaches a preset number $S$. That is, we memorize $S$ features for each category. We provide the pseudo code in the supplementary material.

**Training matrices.** Memorized features, extracted in the previous stage $t-1$, are not compatible with a current classifier $w^t$. To address this, the work of [19] in image classification proposes to exploit two-layer perceptrons, called FAN, where the number of parameters is roughly $32D^2$. Two main limitations of FAN are as follows: (1) It updates all features without considering categories. Adopting separate FANs for each category alleviates this issue, but it is computationally expensive. (2) FAN ignores the relations between memorized features. The structural information in the feature space is crucial for the generalization ability of classifiers. To address these problems, we propose to train category-specific rotation matrices to update features of each category separately. The rotation transform enables maintaining the relations between features within the same category, while the number of parameters for each matrix is $0.5(D^2 - D)$ only (See the supplementary material). To this end, we first define a skew-symmetric matrix $\mathbf{S}_c$ of size $D \times D$ for the category $c \in C_{\text{prev}}^t$. A rotation matrix for the category $c$ is then defined using the Cayley transform as follows:

$$\mathbf{R}_c = (\mathbf{I} - \mathbf{S}_c)(\mathbf{I} + \mathbf{S}_c)^{-1}, \tag{11}$$

where $\mathbf{I}$ indicates an identity matrix. Learning the parameters for the matrices is challenging, since training samples of $D^t$ are labeled only for $C_{\text{new}}^t$. To handle this, we extract features $f^{t-1}$ from a previous feature extractor $\phi^{t-1}$, and compute a correlation score as follows:

$$v_c(\mathbf{p}) = \sum_{s=1}^{S} \text{ReLU}\left(\frac{f^{t-1}(\mathbf{p})}{\|f^{t-1}(\mathbf{p})\|} \cdot \frac{m_c(s)}{\|m_c(s)\|}\right), \tag{12}$$

where we denote by $m_c(s) \in \mathbb{R}^D$ the $s$-th item in memorized features of the category $c$. This allows to identify features associated with previous categories $C_{\text{prev}}^t$, since $f^{t-1}$ and $m_c(s)$ share the same feature space. We then apply the softmax function over the correlation score as follows:

$$\sigma_c(\mathbf{p}) = \frac{e^{\tau v_c(\mathbf{p})}}{\sum_{\mathbf{p}} e^{\tau v_c(\mathbf{p})}}, \tag{13}$$

where $\tau$ is a temperature controlling the sharpness of $\sigma_c$. Note that both features $f^{t-1}$ and $f^t$ at position $\mathbf{p}$ encode the same semantic information. Using this fact, we define prototypes of the category $c$ for previous and current stages, $r_c^{t-1}$ and $r_c^t$, respectively, by computing a weighted average of each feature map, as follows:

$$r_c^{t-1} = \sum_{\mathbf{p}} \sigma_c(\mathbf{p}) f^{t-1}(\mathbf{p}), \quad r_c^t = \sum_{\mathbf{p}} \sigma_c(\mathbf{p}) f^t(\mathbf{p}). \tag{14}$$

We can match the prototypes for each category $c$ via the corresponding rotation matrix $\mathbf{R}_c$. Namely, each matrix $\mathbf{R}_c$ rotates a previous prototype $r_c^{t-1}$ to align it with a current one $r_c^t$. To train the matrix $\mathbf{R}_c$, we define an objective function as follows:

$$L_{\text{S2}} = \lambda_{\text{ROT}} L_{\text{FID}} + (1 - \lambda_{\text{ROT}}) L_{\text{REG}}, \tag{15}$$

where we denote by $L_{\text{FID}}$ and $L_{\text{REG}}$ fidelity and regularization terms, respectively, balanced by the parameter $\lambda_{\text{ROT}}$. The fidelity term maximizes cosine similarity as follows:

$$L_{\text{FID}} = \sum_{c \in C_{\text{prev}}^t} \left( 1 - \frac{\hat{r}_c}{\|\hat{r}_c\|} \cdot \frac{r_c^t}{\|r_c^t\|} \right), \tag{16}$$

where $\hat{r}_c = \mathbf{R}_c r_c^{t-1}$. This encourages the matrices $\mathbf{R}_c$ to align $\hat{r}_c$ with $r_c^t$. The regularization term enforces the rotated prototypes $\hat{r}_c$ to be compatible with the current classifier $w^t$ to better guide the alignment process. To this end, we compute a CE loss using the softmax classifier $w^t$ as follows:

$$L_{\text{REG}} = \sum_{c \in C_{\text{prev}}^t} -\log \left( \frac{e^{\hat{r}_c \cdot w_c^t}}{\sum_{i \in C_{\text{all}}^t} e^{\hat{r}_c \cdot w_i^t}} \right). \tag{17}$$

### 3.4  Step 3

In the third step, we first update memorized features of each category as follows:

$$\hat{m}_c(s) = \mathbf{R}_c m_c(s). \tag{18}$$

The updated features along with training samples of $D^t$ are then used to fine-tune a classifier $w^t$ with the following objective:

$$L_{\text{S3}}(\mathbf{p}) = L_{\text{FL}}(\mathbf{p}) + \lambda_{\text{ALI}} L_{\text{ALI}}(\mathbf{p}) + \lambda_{\text{MEM}} L_{\text{MEM}}. \tag{19}$$

We denote by $L_{\text{FL}}$ and $L_{\text{MEM}}$ focal loss (FL) [24] and CE terms, respectively, defined as follows:

$$L_{\text{FL}}(\mathbf{p}) = -(1 - p_{\hat{c}}^t(\mathbf{p}))^\alpha \log p_{\hat{c}}^t(\mathbf{p}), \quad \hat{c} = \begin{cases} y(\mathbf{p}), & \mathbf{p} \in \mathcal{R}_{\text{new}}^t \\ \operatorname{argmax}_{k \in C_{\text{prev}}^t} p_k^{t-1}(\mathbf{p}), & \mathbf{p} \notin \mathcal{R}_{\text{new}}^t \end{cases}, \tag{20}$$

and

$$L_{\text{MEM}} = \sum_{c \in C_{\text{prev}}^t} \sum_{s=1}^{S} -\log \left( \frac{e^{\hat{m}_c(s) \cdot w_c^t}}{\sum_{k \in C_{\text{all}}^t} e^{\hat{m}_k(s) \cdot w_k^t}} \right), \tag{21}$$

where we set $\alpha$ to 2 by default. Following [3, 9, 26, 30], we mark unlabeled regions in the training samples of $D^t$ as predictions obtained from a previous model on-the-fly. Note that we freeze a feature extractor and the rotation matrices for fine-tuning the classifier.

## 4  Experiments

In this section, we present a quantitative comparison between our method and the state of the art, and show ablation studies. More results including qualitative comparisons can be found in the supplementary material.

Table 3: Quantitative results on ADE20K [40] in terms of IoU scores. SSUL-M [3] uses a replay buffer that consists of 300 previously seen images together with corresponding ground-truth labels. Numbers in bold are the best performance, while underlined ones are the second best. We show standard deviations in parentheses. Numbers for other methods [2, 29] are taken from SSUL. †: Results are obtained by running the source codes provided by the authors.

| Methods | 100-50(1) | | | | 50-100(2) | | | | 100-50(5) | | | |
|---|---|---|---|---|---|---|---|---|---|---|---|---|
| | $mIoU_{base}$ | $mIoU_{new}$ | mIoU | hIoU | $mIoU_{base}$ | $mIoU_{new}$ | mIoU | hIoU | $mIoU_{base}$ | $mIoU_{new}$ | mIoU | hIoU |
| Without memorized images or features | | | | | | | | | | | | |
| ILT [29] | 18.29 | 14.40 | 17.00 | 16.11 | 3.53 | 12.85 | 9.70 | 5.54 | 0.08 | 1.31 | 0.49 | 0.15 |
| MiB [2] | 40.52 | 17.17 | 32.79 | 24.12 | 45.57 | 21.01 | 29.31 | 28.76 | 38.21 | 11.12 | 29.24 | 17.23 |
| PLOP† [9] | 42.10 (0.02) | 16.22 (0.15) | 33.53 (0.06) | 23.42 (0.15) | 48.24 (0.03) | 21.31 (0.08) | 30.40 (0.06) | 29.56 (0.08) | 40.78 (0.04) | 14.02 (0.03) | 31.92 (0.04) | 20.87 (0.03) |
| SSUL [3] | 41.28 | 18.02 | 33.58 | 25.09 | 48.38 | 20.15 | 29.56 | 28.45 | 40.20 | 18.75 | 33.10 | 25.57 |
| ALIFE | 42.18 (0.08) | 23.07 (0.51) | 35.86 (0.12) | 29.83 (0.41) | 48.98 (0.12) | 25.69 (0.20) | 33.56 (0.16) | 33.70 (0.19) | 41.02 (0.23) | 22.76 (0.55) | 34.98 (0.34) | 29.28 (0.51) |
| With memorized images or features | | | | | | | | | | | | |
| SSUL-M [3] | 42.79 | 17.54 | 34.37 | 24.88 | 49.12 | 20.10 | 29.77 | 28.53 | 42.86 | 17.66 | 34.46 | 25.01 |
| ALIFE-M | 42.28 (0.05) | 23.58 (0.45) | 36.09 (0.12) | 30.28 (0.36) | 48.99 (0.07) | 26.15 (0.10) | 33.87 (0.09) | 34.10 (0.10) | 41.17 (0.21) | 23.07 (0.22) | 35.18 (0.21) | 29.57 (0.23) |

## 4.1 Implementation details

**Dataset and evaluation.** We evaluate our method on standard ISS benchmarks (PASCAL VOC [10] and ADE20K [40]). PASCAL VOC provides $10,582$ training [13] and $1,449$ validation samples with 20 object and one background categories, while ADE20K consists of $20,210$ and $2,000$ samples for training and validation, respectively, with 150 object/stuff categories. There are three incremental scenarios for each dataset, where we denote by each scenario $A$-$B(C)$. $A$, $B$ and $C$ indicate the number of categories at a base stage, the total number of novel categories, and the number of incremental stages, respectively. We follow the same scenarios as in [2, 3, 9, 26, 30], unless otherwise specified. Following the common practice [3, 9], we focus on an overlapped setting, where unlabeled regions could contain either previous or future categories. For evaluation, we report IoU scores on the validation set for each dataset, and do not exploit test-time augmentation or dense CRF techniques [22]. We denote by $mIoU_{base}$, $mIoU_{new}$, and mIoU the mean IoU scores over base, new, and all categories, respectively. Computing an IoU score over all categories (*i.e.* mIoU) is typical, but we have found that this does not reflect IoU scores for new categories well. To address this, as in the evaluation protocol of zero-shot learning methods (*e.g.*, [37]), we propose to use the harmonic mean (hIoU) of $mIoU_{base}$ and $mIoU_{new}$. For all experiments, we report scores averaged over 3 runs (*i.e.*, different random seeds).

**Training.** Following [2, 3, 9, 26, 30], we adopt DeepLab-V3 [5] with ResNet-101 [14]. ResNet-101 is initialized with pre-trained weights for classification on ImageNet [7]. We use the SGD optimizer with an initial learning rate set to 1e-2 and 1e-3 for base and incremental stages, respectively. DeepLab-V3 is trained for 30 and 60 epochs at a base stage ($t = 1$) on PASCAL VOC [10] and ADE20K [40], respectively. For each incremental stage ($t > 1$), we perform a cross-validation to choose the number of epochs on PASCAL VOC, while fixing it to 60 on ADE20K. We train rotation matrices for 10 epochs using the Adam optimizer with an initial learning rate of 1e-3, and fix a preset number $S$ and a temperature value $\tau$ to $1,000$ and 10 for all experiments. We fine-tune a classifier for 1 epoch using the SGD optimizer with an initial learning rate of 1e-3. For all experiments, we adjust the learning rate by the poly schedule. We provide a detailed description of hyperparameter settings in the supplementary material.

## 4.2 Comparison with the state of the art

**ADE20K.** We compare in Table 3 our approach with state-of-the-art methods, including MiB [2], PLOP [9], and SSUL [3]. Note that RECALL [26] is not designed to handle stuff categories, and results on ADE20K [40] are not available. From this table, we have three findings as follows: (1) Our approach exploiting the first step only, denoted by ALIFE, already outperforms all other methods in terms of both mIoU and hIoU scores by significant margins for all scenarios. This validates the effectiveness of our approach without memorizing features. In particular, we can see that ALIFE even

Table 4: Quantitative results on PASCAL VOC [10] in terms of IoU scores. SSUL-M [3] memorizes 100 images in total, while RECALL [26] uses 500 images for each previous category. Note that both SSUL and SSUL-M also require an off-the-shelf saliency detector [16] on PASCAL VOC. Numbers in bold are the best performance, while underlined ones are the second best. We show standard deviations in parentheses. Numbers for other methods are taken from corresponding papers. †: Results are obtained with the source codes provided by the authors.

| Methods | 20-1(1) | | | | 16-5(1) | | | | 16-5(5) | | | |
|---|---|---|---|---|---|---|---|---|---|---|---|---|
| | $mIoU_{base}$ | $mIoU_{new}$ | mIoU | hIoU | $mIoU_{base}$ | $mIoU_{new}$ | mIoU | hIoU | $mIoU_{base}$ | $mIoU_{new}$ | mIoU | hIoU |
| *Without memorized images or features* | | | | | | | | | | | | |
| EWC [21] | 26.90 | 14.00 | 26.30 | 18.42 | 24.30 | 35.50 | 27.10 | 28.85 | 0.30 | 4.30 | 1.30 | 0.56 |
| LwF-MC [23] | 64.40 | 13.30 | 61.90 | 22.05 | 58.10 | 35.00 | 52.30 | 43.68 | 6.40 | 8.40 | 6.90 | 7.26 |
| ILT [29] | 67.75 | 10.88 | 65.05 | 18.75 | 67.08 | 39.23 | 60.45 | 49.51 | 8.75 | 7.99 | 8.56 | 8.35 |
| MiB† [2] | 70.42 (0.13) | 17.70 (1.89) | 67.91 (0.17) | 28.25 (2.44) | 76.68 (0.11) | 49.03 (0.27) | 70.09 (0.14) | 59.81 (0.23) | 37.98 (0.72) | 12.28 (0.19) | 31.86 (0.57) | 18.56 (0.28) |
| SDR [30] | 71.30 | 23.40 | 69.00 | 35.24 | 76.30 | 50.20 | 70.10 | 60.56 | 47.30 | 14.70 | 39.50 | 22.43 |
| PLOP† [9] | 75.89 (0.21) | 34.90 (0.93) | 73.94 (0.24) | 47.81 (0.91) | 76.37 (0.14) | 49.55 (0.29) | 69.98 (0.18) | 60.10 (0.26) | 64.51 (0.13) | 19.93 (0.20) | 53.90 (0.06) | 30.45 (0.21) |
| SSUL [3] | **77.73** | 29.68 | **75.44** | 42.96 | **77.82** | 50.10 | 71.22 | 60.96 | **77.31** | 36.59 | **67.61** | **49.67** |
| ALIFE | 76.61 (0.52) | **49.36** (1.01) | 75.31 (0.52) | **60.03** (0.84) | 77.18 (0.66) | **52.52** (0.48) | **71.31** (0.52) | **62.50** (0.41) | 64.44 (1.24) | 34.91 (1.05) | 57.41 (1.19) | 45.29 (1.18) |
| *With memorized images or features* | | | | | | | | | | | | |
| SSUL-M [3] | **78.83** | 49.76 | **76.49** | 61.01 | **78.40** | 55.80 | **73.02** | **65.20** | **78.36** | 49.01 | **71.37** | **60.30** |
| RECALL [26] | 68.10 | **55.30** | 68.60 | 61.04 | 67.70 | 54.30 | 65.60 | 60.26 | 67.80 | **50.90** | 64.80 | 58.15 |
| ALIFE-M | 76.72 (0.57) | 52.29 (0.62) | 75.56 (0.55) | **62.19** (0.49) | 77.66 (0.36) | 55.27 (0.98) | 72.33 (0.21) | 64.57 (0.59) | 66.09 (0.64) | 38.81 (1.86) | 59.59 (0.93) | 48.89 (1.64) |

outperforms SSUL-M [3] that memorizes 300 images along with ground-truth labels for replaying. A plausible reason is that SSUL freezes a feature extractor, limiting the flexibility to deal with new categories. (2) ALIFE shows substantial IoU gains over MiB [2] using CCE and CKD for all scenarios. This verifies that both CCE and CKD are not always helpful for ISS. ALI is free from the limitations of CCE and CKD, and it allows our model to better learn new categories without forgetting previous ones. (3) Our approach memorizing features, denoted by ALIFE-M, improves the performance over ALIFE in terms of all metrics for all scenarios. Note that SSUL-M even performs worse than SSUL for 100-50(1) and 100-50(6) cases. Considering that we rely on at least 9 times less memory requirements than SSUL-M for 100-50(1) and 50-100(2) cases, the gains from memorizing features are remarkable compared to those of SSUL-M over SSUL.

**PASCAL VOC.** We show in Table 4 quantitative results on PASCAL VOC [10]. Note that a comparison of SSUL [3] (SSUL-M) and other methods including ours is not fair, since it additionally exploits an off-the-shelf saliency detector [16] that brings significant improvements on PASCAL VOC. From this table, we can see that ALIFE outperforms all other methods in terms of hIoU scores for 20-1(1) and 16-5(1) scenarios, further demonstrating the effectiveness of our approach without memorizing features. We can also see that ALIFE-M gives substantial gains over ALIFE in terms of all metrics for all scenarios. In particular, the largest mIoU and hIoU gains of 2.18% and 3.60%, respectively, are reported on 16-5(5). This confirms once again that our feature replay scheme is effective even for the most challenging scenario on PASCAL VOC. SSUL-M [3] and RECALL [26] largely outperform ours only for 16-5(5), but they require at least 604 and 15 times more memory footprint than ALIFE-M, respectively. Moreover, RECALL has difficulty for handling stuff categories, and SSUL performs poorly on ADE20K [40], where the saliency detector is not applicable. Our approach is versatile in that it is free to handle stuff categories without performance degradation on ADE20K (See Table 3).

### 4.3 Discussion

**The ordering of categories.** To show the robustness of our approach to different category orderings, we generate two random category orderings for 16-5(1) and 20-1(1) cases, and report the mean and standard deviation of hIoU scores over three orderings: one alphabetical and two random orderings. We compare in Fig. 2 our approach without memorizing fea-

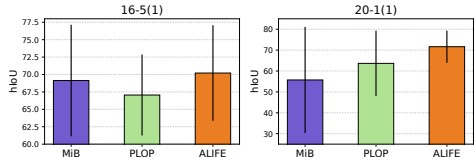

Figure 2: Comparison of average hIoU scores over three different category orderings on PASCAL VOC [10].

tures, denoted by ALIFE, to MiB [2] and PLOP [9]. Results of MiB and PLOP are obtained by running the source codes provided by the authors. We can clearly see that ALIFE produces better results than other methods on both 16-5(1) and 20-1(1) cases.

**Ablation study on the first step.** We show in Table 5 an ablation analysis on different loss terms of our approach without memorizing features (Eq. (10)). The first row shows that using the CE term alone performs poorly due to catastrophic forgetting. We can see from the second row that ALI alleviates the catastrophic forgetting problem remarkably. The third row shows that additionally using the KD term for labeled regions on top of CE and ALI terms further gives IoU gains. On the other hand, we can see from the fourth

Table 5: Comparison of IoU scores using different loss terms of our approach on 16-5(1) of PASCAL VOC [10]. *Labeled*, *Unlabeled*, and *All* indicate that KD is applied for labeled, unlabeled, and all regions, respectively.

| CE | ALI | KD | | | $\text{mIoU}_{\text{base}}$ | $\text{mIoU}_{\text{new}}$ | mIoU | hIoU |
|----|-----|----|----|----|----|----|----|----|
| | | *Labeled* | *Unlabeled* | *All* | | | | |
| ✓ | | | | | 16.45 | 6.94 | 14.19 | 9.74 |
| ✓ | ✓ | | | | 75.50 | 49.81 | 69.39 | 60.02 |
| ✓ | ✓ | ✓ | | | **77.18** | **52.52** | **71.31** | **62.50** |
| ✓ | ✓ | | ✓ | | 75.45 | 49.75 | 69.33 | 59.97 |
| ✓ | ✓ | | | ✓ | 76.25 | 50.99 | 70.24 | 61.12 |

row that applying the KD term for unlabeled regions rather degrades the performance. Since our ALI already encourages a current model to imitate knowledge of a previous one for unlabeled regions, we conjecture that additionally applying the KD term for those regions is redundant. The last row shows that using the KD term for all regions is beneficial to improving the performance slightly.

**Ablation study on the third step.** We report in Table 6 IoU scores for different losses of our method in the third step (Eq. (19)). From the first four rows, we can see that 1) using CE or FL terms alone degrade the performance. This is because training samples of a current dataset mainly contain new categories, causing the class imbalance between previous and new categories; 2) FL provides better results than CE; 3) The pseudo labeling strategy works favorably for both CE and FL, mitigating the imbalance. From the last three rows, we can see that 1) replaying 1K features for each previous category lessens the influence

Table 6: Comparison of IoU scores using different loss terms of our approach on 16-5(1) of PASCAL VOC [10]. *Labeled*: CE or FL is applied only for labeled regions. *All\**: To apply CE or FL for all regions, we mark unlabeled regions as predictions of a previous model on-the-fly.

| CE | | FL | | ALI | MEM | $\text{mIoU}_{\text{base}}$ | $\text{mIoU}_{\text{new}}$ | mIoU | hIoU |
|----|----|----|----|----|----|----|----|----|----|
| *Labeled* | *All\** | *Labeled* | *All\** | | | | | | |
| ✓ | | | | | | 76.11 | 48.03 | 69.42 | 58.89 |
| | ✓ | | | | | 76.71 | 50.37 | 70.44 | 60.81 |
| | | ✓ | | | | 76.51 | 49.70 | 70.13 | 60.26 |
| | | | ✓ | | | 76.81 | 50.98 | 70.66 | 61.29 |
| | | | ✓ | | ✓ | 77.24 | 54.90 | 71.92 | 64.17 |
| | | | ✓ | ✓ | | 77.25 | 52.88 | 71.44 | 62.78 |
| | | | ✓ | ✓ | ✓ | **77.66** | **55.27** | **72.33** | **64.57** |

of the class imbalance problem significantly and 2) ALI is also helpful to alleviate the imbalance problem, further boosting the performance.

**Limitation.** Our approach to memorizing features for an experience replay reduces memory requirements significantly and avoids data privacy issues. Nonetheless, it requires more memory than other methods [2, 9, 30] that do not adopt the experience replay. Since our approach to using ALI without memorizing features shows state-of-the-art results on standard ISS benchmarks [10, 40], we believe that ALI could give useful insights for developing ISS methods that do not rely on the replay strategy. It is also worth noting that all existing methods including ours assume that newly incoming samples are clean and reliable. However, in practice, training samples of a new task might be biased and unreliable. This raises new concerns: 1) addressing noisy samples during training and 2) memorizing clean samples only. Handling these potential risks would also be an interesting future direction for ISS.

## 5 Conclusion

We have introduced a new ISS method, ALIFE, that alleviates catastrophic forgetting and reduces memory requirements for an experience replay. First, we have presented a gradient analysis of CCE and CKD for better understanding the effects on catastrophic forgetting, and have proposed ALI incorporating the merits of CCE and CKD. Second, we have proposed a feature replay scheme using the Cayley transform that requires less memory footprint than memorizing raw images. Finally, we have shown that ALIFE sets a new state of the art on standard ISS benchmarks.

## Acknowledgments and Disclosure of Funding

This work was supported by Institute of Information & Communications Technology Planing & Evaluation (IITP) grant funded by the Korea government (MSIT) (No.RS-2022-00143524, Development of Fundamental Technology and Integrated Solution for Next-Generation Automatic Artificial Intelligence System) and the Yonsei Signature Research Cluster Program of 2022 (2022-22-0002).

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
