# ALIFE: Adaptive Logit Regularizer and Feature Replay for Incremental Semantic Segmentation Supplement

**Youngmin Oh**     **Donghyeon Baek**     **Bumsub Ham**[*]
School of Electrical and Electronic Engineering, Yonsei University
https://cvlab.yonsei.ac.kr/projects/ALIFE

Here we show detailed derivations together with a further analysis of CCE and CKD (Sec. A). We describe more details for experimental settings and hyperparamters, and present a pseudo code of our approach (Sec. B). We also provide more discussions on design choices and a feature replay scheme (Sec. C), and show more quantitative and qualitative results (Sec. D).

## A  Derivations together with an analysis of CCE and CKD

**Proposition 1.** *For $c \in C_{\text{prev}}^t$, $q_c^t$ is always larger than $p_c^t$.*

*Proof.*

$$
\begin{aligned}
\frac{q_c^t(\mathbf{p})}{p_c^t(\mathbf{p})} &= \frac{e^{z_c^t(\mathbf{p})}}{\sum_{i \in C_{\text{prev}}^t} e^{z_i^t(\mathbf{p})}} \frac{\sum_{j \in C_{\text{all}}^t} e^{z_j^t(\mathbf{p})}}{e^{z_c^t(\mathbf{p})}} \\
&= \frac{\sum_{j \in C_{\text{all}}^t} e^{z_j^t(\mathbf{p})}}{\sum_{i \in C_{\text{prev}}^t} e^{z_i^t(\mathbf{p})}} \\
&= \frac{\sum_{i \in C_{\text{prev}}^t} e^{z_i^t(\mathbf{p})} + \sum_{j \in C_{\text{new}}^t} e^{z_j^t(\mathbf{p})}}{\sum_{i \in C_{\text{prev}}^t} e^{z_i^t(\mathbf{p})}} \qquad\text{(i)} \\
&= 1 + \frac{\sum_{j \in C_{\text{new}}^t} e^{z_j^t(\mathbf{p})}}{\sum_{i \in C_{\text{prev}}^t} e^{z_i^t(\mathbf{p})}} \\
&= 1 + \gamma(\mathbf{p}).
\end{aligned}
$$

Since $\gamma(\mathbf{p})$ is always larger than zero, we can derive the following inequality:

$$
\frac{q_c^t(\mathbf{p})}{p_c^t(\mathbf{p})} = 1 + \gamma(\mathbf{p}) > 1. \qquad\text{(ii)}
$$

□

---

[*]Corresponding author.

36th Conference on Neural Information Processing Systems (NeurIPS 2022).

## A.1 CCE

We first reformulate the CCE loss (Eq. (5) in the main paper) as follows:

$$L_{\text{CCE}}(\mathbf{p}) = \begin{cases} -\log p_{c*}^t(\mathbf{p}), & \mathbf{p} \in \mathcal{R}_{\text{new}}^t \\ -\log p_{\text{cce}}^t(\mathbf{p}), & \mathbf{p} \notin \mathcal{R}_{\text{new}}^t \end{cases}$$

$$= \begin{cases} -z_{c*}^t(\mathbf{p}) \quad + \quad \log\left(\sum_{k \in C_{\text{all}}^t} e^{z_k^t(\mathbf{p})}\right), & \mathbf{p} \in \mathcal{R}_{\text{new}}^t \\ -\log\left(\sum_{i \in C_{\text{prev}}^t} e^{z_i^t(\mathbf{p})}\right) + \log\left(\sum_{k \in C_{\text{all}}^t} e^{z_k^t(\mathbf{p})}\right), & \mathbf{p} \notin \mathcal{R}_{\text{new}}^t \end{cases} \quad \text{(iii)}$$

Then, the gradient of CCE w.r.t $z_c^t$ is computed as follows:

$$\frac{\partial L_{\text{CCE}}(\mathbf{p})}{\partial z_c^t(\mathbf{p})} = \begin{cases} -\mathbb{1}[c = y(\mathbf{p})] \quad + \quad \dfrac{e^{z_c^t(\mathbf{p})}}{\sum_{k \in C_{\text{all}}^t} e^{z_k^t(\mathbf{p})}}, & \mathbf{p} \in \mathcal{R}_{\text{new}}^t \\ -\dfrac{e^{z_c^t(\mathbf{p})}}{\sum_{i \in C_{\text{prev}}^t} e^{z_i^t(\mathbf{p})}} \mathbb{1}[c \in C_{\text{prev}}^t] + \dfrac{e^{z_c^t(\mathbf{p})}}{\sum_{k \in C_{\text{all}}^t} e^{z_k^t(\mathbf{p})}}, & \mathbf{p} \notin \mathcal{R}_{\text{new}}^t \end{cases}$$

$$= \begin{cases} -\mathbb{1}[c = y(\mathbf{p})] \quad + \quad p_c^t(\mathbf{p}), & \mathbf{p} \in \mathcal{R}_{\text{new}}^t \\ -q_c^t(\mathbf{p})\mathbb{1}[c \in C_{\text{prev}}^t] + p_c^t(\mathbf{p}), & \mathbf{p} \notin \mathcal{R}_{\text{new}}^t \end{cases} \quad \text{(iv)}$$

$$= \begin{cases} p_c^t(\mathbf{p}) - 1, & \mathbf{p} \in \mathcal{R}_{\text{new}}^t \ \text{and} \ c = y(\mathbf{p}) \\ p_c^t(\mathbf{p}), & \mathbf{p} \in \mathcal{R}_{\text{new}}^t \ \text{and} \ c \neq y(\mathbf{p}) \\ p_c^t(\mathbf{p}), & \mathbf{p} \notin \mathcal{R}_{\text{new}}^t \ \text{and} \ c \in C_{\text{new}}^t \\ p_c^t(\mathbf{p}) - q_c^t(\mathbf{p}), & \mathbf{p} \notin \mathcal{R}_{\text{new}}^t \ \text{and} \ c \in C_{\text{prev}}^t \end{cases} \quad .$$

We can see that the gradients of Eq. (iv) are the same as those of Table 1 in the main paper. The last row of Table 1 suggests that logit values for all previous categories always increase by $q_c^t - p_c^t$ in the unlabeled regions. We visualize in Fig. A heatmaps of $q_c^t - p_c^t$ for $c \in \{background, chair, person\}$. For visualization, MiB [2] is trained with samples for a tv category after learning 20 categories (*e.g.*, background, chair, and person) on PASCAL VOC [7]. We can see that CCE raises logit values for chair and person categories, even though corresponding ground-truth labels are likely to be the background category. This in turn lessens the discriminability of the model for the previous categories.

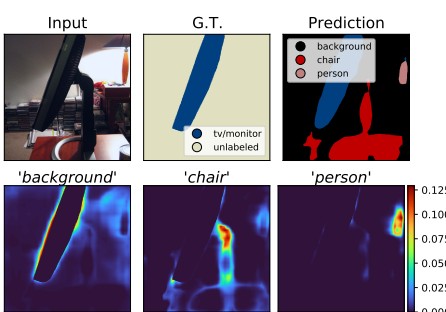

Figure A: The negative effect of CCE on ISS.

## A.2 CKD

The CKD loss (Eq. (6) in the main paper) can be reformulated as follows:

$$L_{\text{CKD}}(\mathbf{p}) = -p_{bg}^{t-1}(\mathbf{p})\log p_{\text{ckd}}^t(\mathbf{p}) + \sum_{k \in C_{\text{prev}}^t \setminus \{bg\}} -p_k^{t-1}(\mathbf{p})\log p_k^t(\mathbf{p})$$

$$= -p_{bg}^{t-1}(\mathbf{p})\log\left(\sum_{i \in \{bg\} \cup C_{\text{new}}^t} e^{z_i^t(\mathbf{p})}\right) + p_{bg}^{t-1}(\mathbf{p})\log\left(\sum_{j \in C_{\text{all}}^t} e^{z_j^t(\mathbf{p})}\right)$$

$$+ \sum_{k \in C_{\text{prev}}^t \setminus \{bg\}}\left(-p_k^{t-1}(\mathbf{p})z_k^t(\mathbf{p}) + p_k^{t-1}(\mathbf{p})\log\left(\sum_{j \in C_{\text{all}}^t} e^{z_j^t(\mathbf{p})}\right)\right)$$

$$= -p_{bg}^{t-1}(\mathbf{p})\log\left(\sum_{i \in \{bg\} \cup C_{\text{new}}^t} e^{z_i^t(\mathbf{p})}\right) + \left(\sum_{k \in C_{\text{prev}}^t} p_k^{t-1}(\mathbf{p})\right) \cdot \log\left(\sum_{j \in C_{\text{all}}^t} e^{z_j^t(\mathbf{p})}\right)$$

$$+ \sum_{k \in C_{\text{prev}}^t \setminus \{bg\}} -p_k^{t-1}(\mathbf{p})z_k^t(\mathbf{p})$$

$$= -p_{bg}^{t-1}(\mathbf{p})\log\left(\sum_{i \in \{bg\} \cup C_{\text{new}}^t} e^{z_i^t(\mathbf{p})}\right) + \log\left(\sum_{j \in C_{\text{all}}^t} e^{z_j^t(\mathbf{p})}\right) + \sum_{k \in C_{\text{prev}}^t \setminus \{bg\}} -p_k^{t-1}(\mathbf{p})z_k^t(\mathbf{p}).$$

(v)

Note that $\sum_{k \in C_{\text{prev}}^t} p_k^{t-1}(\mathbf{p}) = 1$. We then compute the gradient of CKD w.r.t $z_c^t$ as follows:

$$\frac{\partial L_{\text{CKD}}(\mathbf{p})}{\partial z_c^t(\mathbf{p})} = -p_{bg}^{t-1}(\mathbf{p})\frac{e^{z_c^t(\mathbf{p})}}{\sum_{i \in \{bg\} \cup C_{\text{new}}^t} e^{z_i^t(\mathbf{p})}}\mathbb{1}[c \in \{bg\} \cup C_{\text{new}}^t] + \frac{e^{z_c^t(\mathbf{p})}}{\sum_{j \in C_{\text{all}}^t} e^{z_j^t(\mathbf{p})}} - p_c^{t-1}(\mathbf{p})\mathbb{1}[c \in C_{\text{prev}}^t \setminus \{bg\}]$$

$$= -p_{bg}^{t-1}(\mathbf{p})\frac{e^{z_c^t(\mathbf{p})}}{\sum_{i \in \{bg\} \cup C_{\text{new}}^t} e^{z_i^t(\mathbf{p})}}\frac{\sum_{k \in C_{\text{all}}^t} e^{z_k^t(\mathbf{p})}}{\sum_{k \in C_{\text{all}}^t} e^{z_k^t(\mathbf{p})}}\mathbb{1}[c \in \{bg\} \cup C_{\text{new}}^t] + p_c^t(\mathbf{p}) - p_c^{t-1}(\mathbf{p})\mathbb{1}[c \in C_{\text{prev}}^t \setminus \{bg\}]$$

$$= -p_{bg}^{t-1}(\mathbf{p})\frac{p_c^t(\mathbf{p})}{\sum_{i \in \{bg\} \cup C_{\text{new}}^t} p_i^t(\mathbf{p})}\mathbb{1}[c \in \{bg\} \cup C_{\text{new}}^t] + p_c^t(\mathbf{p}) - p_c^{t-1}(\mathbf{p})\mathbb{1}[c \in C_{\text{prev}}^t \setminus \{bg\}]$$

$$= -p_{bg}^{t-1}(\mathbf{p})\frac{p_c^t(\mathbf{p})}{p_{\text{ckd}}^t(\mathbf{p})}\mathbb{1}[c \in \{bg\} \cup C_{\text{new}}^t] + p_c^t(\mathbf{p}) - p_c^{t-1}(\mathbf{p})\mathbb{1}[c \in C_{\text{prev}}^t \setminus \{bg\}]$$

$$= \begin{cases} p_c^t(\mathbf{p}) - p_c^{t-1}(\mathbf{p}), & c \in C_{\text{prev}}^t \setminus \{bg\} \\ p_c^t(\mathbf{p}) - p_{bg}^{t-1}(\mathbf{p})\dfrac{p_c^t(\mathbf{p})}{p_{\text{ckd}}^t(\mathbf{p})}, & c \in \{bg\} \cup C_{\text{new}}^t \end{cases}$$

$$= \begin{cases} p_c^t(\mathbf{p}) - p_c^{t-1}(\mathbf{p}), & c \in C_{\text{prev}}^t \setminus \{bg\} \\ \left(1 - \dfrac{p_{bg}^{t-1}(\mathbf{p})}{p_{\text{ckd}}^t(\mathbf{p})}\right)p_c^t(\mathbf{p}), & c \in \{bg\} \cup C_{\text{new}}^t \end{cases}$$

$$= \begin{cases} p_c^t(\mathbf{p}) - p_c^{t-1}(\mathbf{p}), & c \in C_{\text{prev}}^t \setminus \{bg\} \\ \left(p_{\text{ckd}}^t(\mathbf{p}) - p_{bg}^{t-1}(\mathbf{p})\right)\dfrac{p_c^t(\mathbf{p})}{p_{\text{ckd}}^t(\mathbf{p})}, & c \in \{bg\} \cup C_{\text{new}}^t \end{cases}.$$

(vi)

We can see that $p_c^t - p_c^{t-1}$ and $(p_{ckd}^t - p_{bg}^{t-1})\frac{p_c^t}{p_{ckd}^t}$ in Eq. (vi) are the same as the gradients of CKD in the main paper (See Eqs. (7) and (8)).

To further compare KD and CKD, we first assume that a previous model outputs confident predictions. Namely, probabilities of the previous model show one clear peak as follows:

$$p_{\hat{y}}^{t-1}(\mathbf{p}) \approx 1, \qquad \text{(vii)}$$

where $\hat{y} = \text{argmax}_{k \in C_{\text{prev}}^t} p_k^{t-1}(\mathbf{p})$. We empirically show that this assumption is valid in Fig. B. We then simplify KD and CKD terms (See Eqs. (4) and (6) in the main paper) as follows:

$$L_{\text{KD}}(\mathbf{p}) \approx -p_{\hat{y}}^{t-1}(\mathbf{p}) \log q_{\hat{y}}^t(\mathbf{p}) \quad \text{and} \quad L_{\text{CKD}}(\mathbf{p}) \approx \begin{cases} -p_{bg}^{t-1}(\mathbf{p}) \log p_{ckd}^t(\mathbf{p}), & \hat{y} \in \{bg\} \\ -p_{\hat{y}}^{t-1}(\mathbf{p}) \log p_{\hat{y}}^t(\mathbf{p}), & \hat{y} \in C_{\text{prev}}^t \setminus \{bg\} \end{cases}.$$

$$\text{(viii)}$$

We compute gradients of each term w.r.t $z_{\hat{y}}^t$ for $\hat{y} \in C_{\text{prev}}^t \setminus \{bg\}$ as follows:

$$\frac{\partial L_{\text{KD}}(\mathbf{p})}{\partial z_{\hat{y}}^t(\mathbf{p})} \approx q_{\hat{y}}^t(\mathbf{p}) - p_{\hat{y}}^{t-1}(\mathbf{p}) = \alpha_{\text{kd}}(\mathbf{p}) \quad \text{and} \quad \frac{\partial L_{\text{CKD}}(\mathbf{p})}{\partial z_{\hat{y}}^t(\mathbf{p})} \approx p_{\hat{y}}^t(\mathbf{p}) - p_{\hat{y}}^{t-1}(\mathbf{p}) = \alpha_{\text{ckd}}(\mathbf{p}). \quad \text{(ix)}$$

Table A summarizes the comparison between KD and CKD into three cases. In the case of ①, a current model produces the probability $p_{\hat{y}}^t$ lower than the target one $p_{\hat{y}}^{t-1}$. Thus, $z_{\hat{y}}^t$ should increase in order that $p_{\hat{y}}^t$ follows $p_{\hat{y}}^{t-1}$. Although both KD and CKD raise $z_{\hat{y}}^t$, we can see that CKD raises $z_{\hat{y}}^t$ more strongly than its counterpart ($|\alpha_{\text{kd}}| < |\alpha_{\text{ckd}}|$). This suggests that CKD helps the current model to produce $p_{\hat{y}}^t$ similar to $p_{\hat{y}}^{t-1}$ more quickly.

Table A: Comparison of gradients of KD and CKD w.r.t $z_{\hat{y}}^t$ for $\hat{y} \in C_{\text{prev}}^t \setminus \{bg\}$.

| case ① $(p_{\hat{y}}^{t-1} \geq q_{\hat{y}}^t > p_{\hat{y}}^t)$ | $0 \geq \alpha_{\text{kd}} > \alpha_{\text{ckd}}$ |
| --- | --- |
| case ② $(q_{\hat{y}}^t > p_{\hat{y}}^{t-1} \geq p_{\hat{y}}^t)$ | $\alpha_{\text{kd}} > 0 \geq \alpha_{\text{ckd}}$ |
| case ③ $(q_{\hat{y}}^t > p_{\hat{y}}^t \geq p_{\hat{y}}^{t-1})$ | $\alpha_{\text{kd}} > \alpha_{\text{ckd}} \geq 0$ |

In the case of ②, the current model outputs $p_{\hat{y}}^t$ lower than $p_{\hat{y}}^{t-1}$, indicating that $z_{\hat{y}}^t$ needs to increase. However, note that the probability $q_{\hat{y}}^t$ computed without considering logit values of new categories is larger than the target one $p_{\hat{y}}^{t-1}$ (*i.e.*, $\alpha_{\text{kd}} > 0$). Hence, KD rather reduces $z_{\hat{y}}^t$, which results in reducing $p_{\hat{y}}^t$. On the other hand, CKD raises $z_{\hat{y}}^t$ in order that $p_{\hat{y}}^t$ reaches $p_{\hat{y}}^{t-1}$. In the case of ③, the current model already produces the probability $p_{\hat{y}}^t$ larger than the target one $p_{\hat{y}}^{t-1}$. In this case, it is unnecessary to reduce the logit value $z_{\hat{y}}^t$, since reducing $z_{\hat{y}}^t$ could make the probability $p_{\hat{y}}^t$ lower than the target one $p_{\hat{y}}^{t-1}$. KD however reduces $z_{\hat{y}}^t$ more strongly than CKD ($|\alpha_{\text{kd}}| > |\alpha_{\text{ckd}}|$).

We plot in Fig. B the ratio of each case during training. To this end, we extract feature maps from a mini-batch and compute the number of features belonging to each case. We denote by $N_1$, $N_2$, and $N_3$ the number of features in each case, respectively. Then, the ratio of each case is defined as follows:

$$\text{ratio}_i = \frac{N_i}{N}, \quad i \in \{1, 2, 3\}, \qquad \text{(x)}$$

where $N$ indicates the total number of features, *i.e.*, $N = N_1 + N_2 + N_3$. We compute the ra-

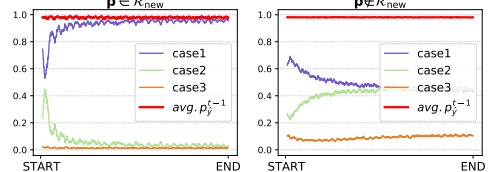

Figure B: Ratio curves of each case in labeled regions (left) and unlabeled ones (right) during training. We also plot the average target probability $p_{\hat{y}}^{t-1}$ (red curves) during training.

tio of each case at every iteration during training. For visualization, we train MiB [2] on 20-1(1) of PASCAL VOC [7]. We have the following observations: (1) We can see that the average target probability $p_{\hat{y}}^{t-1}$ (red curves) is roughly one during training, validating the assumption in Eq. (vii). We can also see that the third case is negligible for both labeled and unlabeled regions; (2) The first case is the most dominant in the labeled regions (See Fig. B left), which is natural in that the current model is likely to output low probabilities $p_{\hat{y}}^t$ for previous categories in those regions[2]. Since CKD raises $z_{\hat{y}}^t$ more strongly than KD in the case of ①, we can interpret that it encourages the current model to imitate the previous one more strongly in the labeled regions. This suggests that

---

[2]Note that the labeled regions contain new categories only.

Table B: Hyperparameter settings. EP: the number of epochs.

| Datasets | Scenarios | $t$ | Step 1 | | | Step 3 | | |
|---|---|---|---|---|---|---|---|---|
| | | | $\lambda_{\mathrm{ALI}}$ | $\lambda_{\mathrm{KD}}$ | EP | $\lambda_{\mathrm{ALI}}$ | $\lambda_{\mathrm{MEM}}$ | EP |
| ADE20K [15] | 100-50(1) | 2 | 1 | 1 | 60 | 0 | 0.5 | 1 |
| | 50-100(2) | 2 | 1 | 20 | 60 | 0 | 0.1 | 1 |
| | | 3 | 1 | 20 | 60 | 0 | 0.5 | 1 |
| | 100-50(5) | 2 | 1 | 1 | 60 | 5 | 2 | 1 |
| | | 3 | 1 | 1 | 60 | 5 | 1 | 1 |
| | | 4 | 1 | 1 | 60 | 2 | 1 | 1 |
| | | 5 | 1 | 1 | 60 | 0 | 0.1 | 1 |
| | | 6 | 1 | 1 | 60 | 0 | 0.5 | 1 |
| PASCAL VOC [7] | 20-1(1) | 2 | 1 | 1 | 5 | 1 | 1 | 1 |
| | 16-5(1) | 2 | 2 | 1 | 10 | 1 | 10 | 1 |
| | 16-5(5) | 2 | 3 | 1 | 10 | 3 | 1 | 1 |
| | | 3 | 5 | 10 | 5 | 5 | 20 | 1 |
| | | 4 | 2 | 1 | 5 | 2 | 1 | 1 |
| | | 5 | 3 | 10 | 5 | 3 | 2 | 1 |
| | | 6 | 2 | 1 | 5 | 1 | 1 | 1 |

CKD acts as a strong regularizer for the current model in those regions; (3) The first two cases are prevalent in the unlabeled regions (See Fig. B right). Note that predictions of the previous model are likely to be correct in unlabeled regions[3]. In this context, it is important for the current model to accurately and quickly imitate such predictions (*i.e.*, $p_{\hat{y}}^{t-1}$) in those regions in order to preserve the knowledge for previous categories. Considering that CKD helps the current model to produce the target probability $p_{\hat{y}}^{t-1}$ more quickly and accurately than KD in the cases of ① and ②, respectively, CKD is more favorable in the unlabeled regions. Note that KD rather prevents the current model from producing the target probability in the case of ②. These empirical studies once again explain the reason why CKD outperforms KD.

## B   More details

**Training rotation matrices.**   Here we provide a detailed description for training rotation matrices. We first define a strictly upper triangular matrix $\mathbf{U}_c$ of size $D \times D$ for a category $c \in C_{\mathrm{prev}}^t$ as follows:

$$\mathbf{U}_c = \begin{pmatrix} 0 & u_{1,2} & u_{1,3} & \dots & u_{1,D} \\ 0 & 0 & u_{2,3} & \dots & u_{2,D} \\ \vdots & & \ddots & & \vdots \\ 0 & 0 & 0 & \dots & u_{D-1,D} \\ 0 & 0 & 0 & \dots & 0 \end{pmatrix}, \tag{xi}$$

where we denote by $u_{i,j}$ the element in the $i$-th row and $j$-th column. Note that the number of non-zero elements for the matrix $\mathbf{U}_c$ is $0.5(D^2 - D)$. Then, the skew-symmetric matrix $\mathbf{S}_c$ can be defined using the corresponding upper triangular matrix $\mathbf{U}_c$ as follows:

$$\mathbf{S}_c = \mathbf{U}_c - \mathbf{U}_c^{\top}. \tag{xii}$$

The skew-symmetric matrix is used to define the rotation matrix $\mathbf{R}_c$ as in Eq. (11) of the main paper. We train the rotation matrices with the objective function in Eq. (15) (See Sec. 3.3 in the main paper). To be specific, the elements of triangular matrices are trained with random initialization. Thus, the number of parameters for each rotation matrix is $0.5(D^2 - D)$ only.

**Pseudo code of our approach.**   We summarize in Algorithm 1 an overall procedure of our approach for incremental stages ($t > 1$). Note that a training process at a base stage ($t = 1$) is equivalent to that of fully-supervised segmentation models.

---

[3]This is because unlabeled regions do not at least contain new categories and the previous model has been trained to classify previous categories.

**Experimental details.** Following the common practice, we have adopted DeepLab-V3 [4] with ResNet-101 [8] pre-trained for ImageNet Classification [5]. In particular, SDR [13] and SSUL [3] use ResNet-101 provided by PyTorch [14], while MiB [2] and PLOP [6] exploit a variant of ResNet-101 using in-place ABN [1] layers. Following SDR and SSUL, we have adopted ResNet-101 provided by PyTorch. For all experiments, we implement our approach using PyTorch, and use two NVIDIA TITAN RTX GPUs along with an Intel i5-9600K CPU. We train DeepLab-V3 and rotation matrices with a batch size of 24, *i.e.*, 12 samples for each GPU.

**Hyperparameter settings.** We empirically set the value of $\tau$ to 10 in order to ensure that correlation scores are sharp enough. For the focal loss [11], we use the default hyperparameters without tuning them. Following the common practice in [2, 3, 6, 12, 13], we perform a cross-validation to choose other hyperparameters, *e.g.*, $\lambda_{\mathrm{ALI}}$, $\lambda_{\mathrm{KD}}$, and $\lambda_{\mathrm{MEM}}$. We summarize in Table B hyperparameters for all experiments. On ADE20K [15], we perform a grid search to set the hyperparameters: $\lambda_{\mathrm{ALI}} \in \{1, 2, 5\}$ and $\lambda_{\mathrm{KD}} \in \{1, 10, 20\}$ for the first step; $\lambda_{\mathrm{ALI}} \in \{0, 1, 2, 5\}$ and $\lambda_{\mathrm{MEM}} \in \{0.1, 0.5, 1, 2\}$ for the third step. In particular, the grid search for the first step is performed only once at the first incremental stage (*i.e.*, $t = 2$), since the search on ADE20K is computationally expensive. In the subsequent stages, we use the same values of hyperparameters. We however perform a grid search at every incremental stage for the third step. This is because we fine-tune a classifier only for a single epoch, which is computationally acceptable. On PASCAL VOC [7], we have empirically found that a hIoU score on the cross-validation set decreases during training, when ISS models are trained with 30 epochs. A plausible reason is that the number of training samples on PASCAL VOC is relatively small. For example, the number of training samples on 20-1(1) of PASCAL VOC is 548, while the number of training samples on 100-50(1) of ADE20K is 9, 390. We thus use a grid search to set the hyperparameters for the first step with $\lambda_{\mathrm{ALI}} \in \{1, 2, 3, 5\}$, $\lambda_{\mathrm{KD}} \in \{1, 10\}$, and the number of epochs (EP) $\in \{5, 10\}$. Since the search cost is relatively mild on PASCAL VOC, we perform the grid search at every incremental stage. For the third step, we also perform a grid search with $\lambda_{\mathrm{ALI}} \in \{1, 2, 3, 5\}$ and $\lambda_{\mathrm{MEM}} \in \{1, 2, 10, 20\}$.

## C More discussions

In this section, we first provide an analysis on ALI. Second, we extend the ablation studies in Tables 5 and 6 of the main paper in order to validate our design choices. Finally, we vary the number of memorized features to analyze its effect on performance, and empirically show that FAN [10] performs poorly when adopted in ISS.

**ALI.** Our ALI can be interpreted as keeping a balance between the maximum logit value and the weighted average of logit values for previous categories adaptively (See Sec.3.2 in the main paper). We conjecture that ALI helps a current model to obtain a more balanced classifier in terms of the norms of classifier weights. To empirically validate this, we show in Fig. C the difference between norms of classifier weights averaged over new and previous categories, denoted by $\|w\|_{\mathrm{new}}$ and $\|w\|_{\mathrm{prev}}$, respectively, during training. For visualization, we train each method on 16-5(1) of PASCAL VOC [7]. From this figure, we can see that applying the CE term alone fails to minimize the difference between $\|w\|_{\mathrm{new}}$ and $\|w\|_{\mathrm{prev}}$ during training. We can also see that both KD and CKD terms are not sufficient to minimize the difference, when being used with the CE term. On the other hand, using the

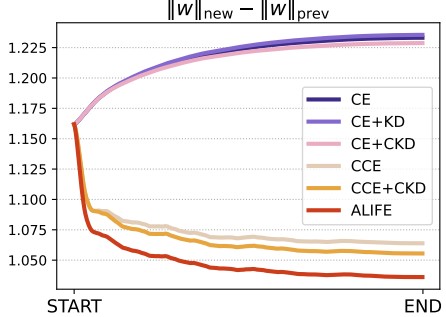

Figure C: $\|w\|_{\mathrm{new}}$ and $\|w\|_{\mathrm{prev}}$ indicate the norms of classifier weights averaged over new and previous categories, respectively. We plot the difference between $\|w\|_{\mathrm{new}}$ and $\|w\|_{\mathrm{prev}}$ during training.

CCE term alone reduces the difference during training, once again verifying that CCE alleviates the overfitting problem. Our approach also minimizes the difference consistently, achieving the lowest difference. This further strengthens the effectiveness of ALI and explains the reason why ALIFE outperforms MiB [2] in Tables 3 and 4 of the main paper.

Table C: Comparison of IoU scores using different loss terms of our approach in the first step. *Labeled*, *Unlabeled*, and *All* indicate that KD is applied for labeled, unlabeled, and all regions, respectively. Numbers in bold are the best performance, while underlined ones are the second best.

| Scenarios | CE | ALI | KD Labeled | KD Unlabeled | KD All | mIoU$_{base}$ | mIoU$_{new}$ | mIoU | hIoU |
|---|---|---|---|---|---|---|---|---|---|
| | ✓ | | | | | 16.45 | 6.94 | 14.19 | 9.74 |
| | ✓ | ✓ | | | | 75.50 | 49.81 | 69.39 | 60.02 |
| **16-5(1)** | ✓ | ✓ | ✓ | | | **77.18** | **52.52** | **71.31** | **62.50** |
| | ✓ | ✓ | | ✓ | | 75.45 | 49.75 | 69.33 | 59.97 |
| | ✓ | ✓ | | | ✓ | 76.25 | 50.99 | 70.24 | 61.12 |
| | ✓ | | | | | 64.63 | 1.11 | 61.61 | 2.19 |
| | ✓ | ✓ | | | | 76.43 | 48.74 | 75.11 | 59.52 |
| **20-1(1)** | ✓ | ✓ | ✓ | | | **76.61** | **49.36** | **75.31** | **60.03** |
| | ✓ | ✓ | | ✓ | | 76.48 | 48.90 | 75.17 | 59.65 |
| | ✓ | ✓ | | | ✓ | 76.51 | 49.07 | 75.20 | 59.79 |

Table D: Comparison of IoU scores using different loss terms of our approach in the third step. *Labeled*: CE or FL is applied only for labeled regions. *All**: To apply CE or FL for all regions, we mark unlabeled regions as predictions of a previous model on-the-fly. Numbers in bold are the best performance, while underlined ones are the second best.

| Scenarios | CE Labeled | CE All* | FL Labeled | FL All* | ALI | MEM | mIoU$_{base}$ | mIoU$_{new}$ | mIoU | hIoU |
|---|---|---|---|---|---|---|---|---|---|---|
| | ✓ | | | | | | 76.11 | 48.03 | 69.42 | 58.89 |
| | | ✓ | | | | | 76.71 | 50.37 | 70.44 | 60.81 |
| | | | ✓ | | | | 76.51 | 49.70 | 70.13 | 60.26 |
| **16-5(1)** | | | | ✓ | | | 76.81 | 50.98 | 70.66 | 61.29 |
| | | | | ✓ | | ✓ | 77.24 | 54.90 | 71.92 | 64.17 |
| | | | | ✓ | ✓ | | 77.25 | 52.88 | 71.44 | 62.78 |
| | | | | ✓ | ✓ | ✓ | **77.66** | **55.27** | **72.33** | **64.57** |
| | ✓ | | | | | | 76.40 | 38.33 | 74.59 | 50.97 |
| | | ✓ | | | | | 76.46 | 46.20 | 75.02 | 57.59 |
| | | | ✓ | | | | 76.38 | 43.27 | 74.80 | 55.22 |
| **20-1(1)** | | | | ✓ | | | 76.42 | 46.63 | 75.01 | 57.91 |
| | | | | ✓ | | ✓ | 76.56 | 47.80 | 75.19 | 58.85 |
| | | | | ✓ | ✓ | | **76.72** | 52.23 | 75.55 | 62.15 |
| | | | | ✓ | ✓ | ✓ | **76.72** | **52.29** | **75.56** | **62.19** |

**More ablation studies.** Here we show more ablation studies including an analysis on our design choices. We report in Table C IoU scores using different loss terms of our approach in the first step. Note that this table contains the results of Table 5 in the main paper. From both scenarios, we can see that additionally using the KD term for labeled regions is beneficial to improving the performance. Table D compares performance using different loss terms of our approach in the third step, while including the results of Table 6 in the main paper. We can see in the first four rows of each scenario that the pseudo labeling strategy improves the performance. In particular, using the FL term along with the pseudo labeling strategy shows decent results. We can also see in the last three rows of each scenario that both ALI and MEM terms bring substantial IoU gains.

**Discussion on a feature replay scheme.** To analyze the effects of the number of memorized features, we vary the preset number $S$, and show in Table E performance w.r.t $S$. From this table, we can see that our approach to memorizing only 100 features for each previous category already shows decent results on both scenarios. The fourth row of each scenario shows that an approach to using features without updating them degrades the performance. This is natural in that the features extracted in the previous stage are not compatible with a classifier at a current stage. We empirically verify in the last row of each scenario that simply adopting FAN [10] in ISS shows sub-optimal performance. For a fair comparison, we train FAN with using Eq. (7) in the paper [10] with carefully tuning hyperparameters. After training, FAN updates saved features to fine-tune a classifier as in Eq. (19) of the main paper.

Table E: Comparison of IoU scores varying the preset number $S$ on PASCAL VOC [7]. No Update: we exploit features extracted from the previous stage to fine-tune a classifier at a current stage. That is, we do not update the features in this case.

| Scenarios | Methods | S | mIoU$_{base}$ | mIoU$_{new}$ | mIoU | hIoU |
|---|---|---|---|---|---|---|
| | | 100 | **77.68** | 55.22 | **72.33** | 64.54 |
| | ALIFE-M | 500 | **77.68** | 55.22 | **72.33** | 64.54 |
| **16-5(1)** | | 1000 | 77.66 | **55.27** | **72.33** | **64.57** |
| | No Update | 1000 | 72.90 | 55.16 | 68.67 | 62.79 |
| | FAN [10] | 1000 | 76.79 | 52.26 | 70.95 | 62.19 |
| | | 100 | 76.70 | 52.28 | 75.54 | 62.18 |
| | ALIFE-M | 500 | **76.72** | **52.29** | **75.56** | **62.19** |
| **20-1(1)** | | 1000 | **76.72** | **52.29** | **75.56** | **62.19** |
| | No Update | 1000 | 76.64 | 52.09 | 75.47 | 62.03 |
| | FAN [10] | 1000 | 74.78 | 49.96 | 73.60 | 59.90 |

Table F: Comparison of IoU scores in the disjoint setting of PASCAL VOC [7]. We show standard deviations in parentheses. All numbers for other methods are copied from SSUL [3]. Numbers in bold are the best performance, while underlined ones are the second best.

| Scenarios | Methods | mIoU$_{base}$ | mIoU$_{new}$ | mIoU | hIoU |
|---|---|---|---|---|---|
| **16-5(1)** | MiB [2] | 71.80 | 43.30 | 64.70 | 54.02 |
| | PLOP [6] | 71.00 | 42.82 | 64.29 | 53.42 |
| | SSUL [3] | **76.44** | **45.60** | **69.10** | **57.12** |
| | ALIFE | 70.66 (0.64) | 44.03 (0.69) | 64.32 (0.42) | 54.25 (0.45) |
| **20-1(1)** | MiB [2] | 69.60 | 25.60 | 67.40 | 37.43 |
| | PLOP [6] | 75.37 | 38.89 | 73.64 | 51.31 |
| | SSUL [3] | **77.38** | 22.43 | 74.76 | 34.78 |
| | ALIFE | 76.31 (0.28) | **50.94** (2.14) | **75.11** (0.35) | **61.08** (1.61) |

**Discussion on the incremental gains from memorizing images or features on ADE20K.** We have observed that the gains from memorizing images or features in Table 3 of the main paper are relatively lower than those in Table 4. We speculate a plausible reason for this issue as follows. It is obvious that ADE20K [15] is more challenging than PASCAL VOC [7]. In particular, ADE20K contains 35 stuff categories, while PASCAL VOC has a single background category. Since 1) the background category always belongs to base categories, and 2) most unlabeled pixels on PASCAL VOC belong to the background category during incremental stages, the unlabeled pixels are less likely to contain future categories even in the overlapped setting. By contrast, on ADE20K, the single background category is split into multiple stuff categories that could possibly belong to future categories. SSUL-M [3] memorizes previously seen images, which contain unlabeled regions, together with ground-truth labels. Since pixels of stuff categories occupy about 60% of all the pixels on ADE20K (See Sec. 4 in [15]), the unlabeled regions could increase when the stuff categories belong to future ones. This might be problematic in that the number of labeled pixels decreases accordingly, lessening the effectiveness of the memoized images. In our case, the feature alignment scheme computes the correlation score between $f^{t-1}(\mathbf{p})$ and $m_c(s)$ (See Eq. (12) in the main paper). If the label of position belongs to future categories, it might result in erroneous correlations, reducing the quality of feature alignment.

## D   More results

**Disjoint setting.** In the disjoint setting, we assume that (1) all categories even including future ones are known in advance and (2) training samples of a current dataset do not contain any categories that would be seen in the future. We report in Table F results of our approach in the disjoint setting of PASCAL VOC [7]. From this table, we can see that ALIFE shows better results than MiB [2] and PLOP [6] in terms of hIoU scores on both 16-5(1) and 20-1(1) cases. In particular, ALIFE outperforms SSUL [3], which exploits an off-the-shelf saliency detector [9], by a large margin in terms of hIoU scores on 20-1(1). This once again demonstrates the effectiveness of our ALI. Note that the disjoint setting is quite different from an overlapped setting in the main paper, where training samples of the current dataset can contain any categories. As pointed out in previous works [2, 3, 6], the overlapped setting is more practical and challenging.

**Qualitative results.** We show in Figs. D and E qualitative results on PASCAL VOC [7]. We show in the last row of each figure failure cases. From the third and fourth columns of Fig. D, we can see that MiB [2] and PLOP [6] struggle to preserve the discriminability for previous categories, *e.g.*, cow, table, and chair. In particular, both methods misclassify cow and chair categories as sheep and sofa, respectively, due to the overfitting problem. ALIFE and ALIFE-M alleviate this problem, showing better segmentation results. We can also see from the last two columns of Fig. D that replaying features gives better results than ALIFE. For example, the fourth row shows that ALIFE-M produces accurate predcitions on the regions for the cow category, where results of ALIFE are incorrect. Figure E also shows a qualitative comparison of ours with MiB and PLOP. Again, the results of MiB and PLOP show that both methods are prone to overfitting to new categories (*e.g.*, a tv category in

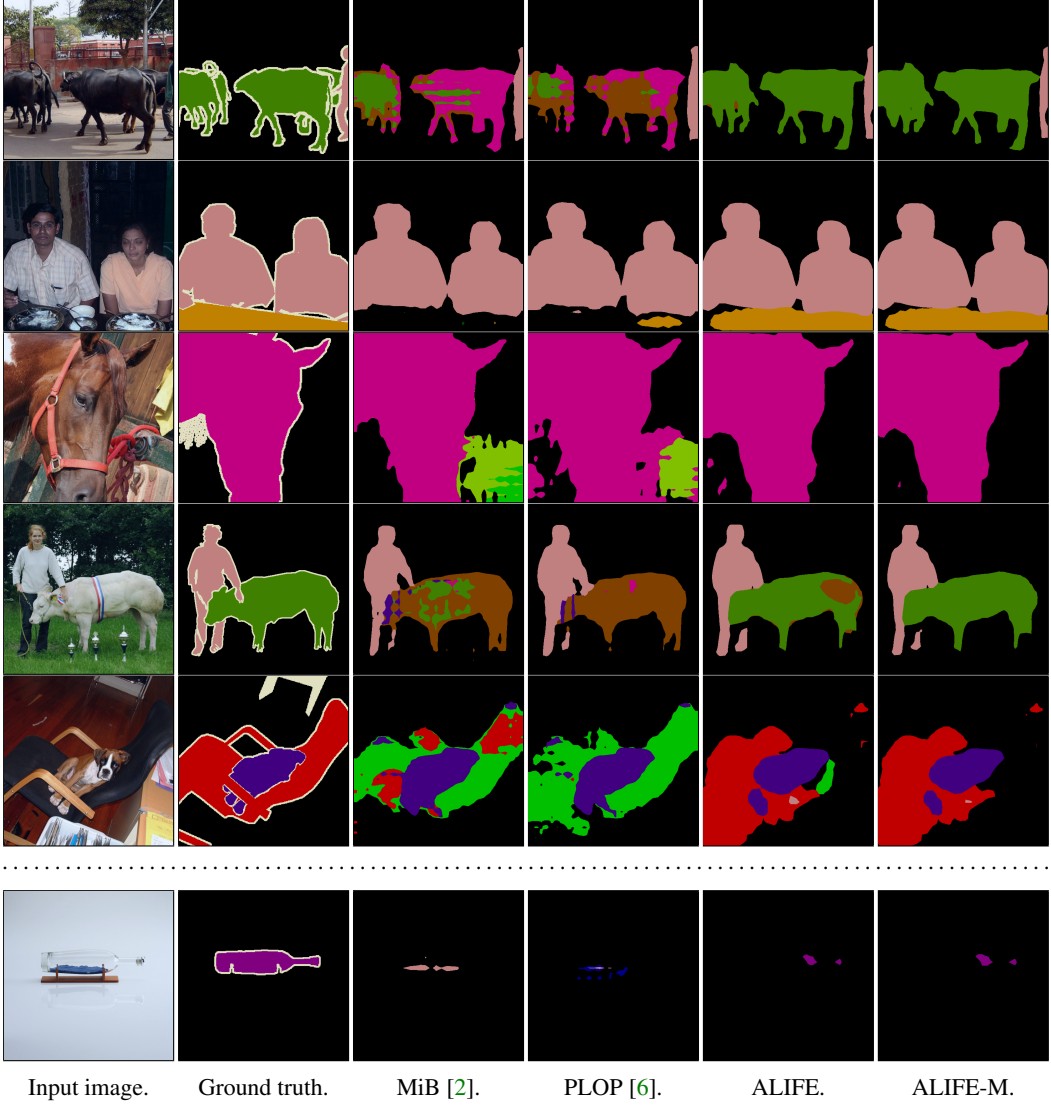

| Input image. | Ground truth. | MiB [2]. | PLOP [6]. | ALIFE. | ALIFE-M. |

Figure D: Visual comparison of ours and other methods [2, 6] on 16-5(1) for the PASCAL VOC [7] validation set. Each method is trained to recognize 5 novel categories (*i.e.* potted plant, sheep, sofa, train, and tv) after learning 16 categories. The last row shows a failure case. Best viewed in color.

this case). We can see that ALIFE already shows decent results and ALIFE-M further improves the quality of segmentation results.

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

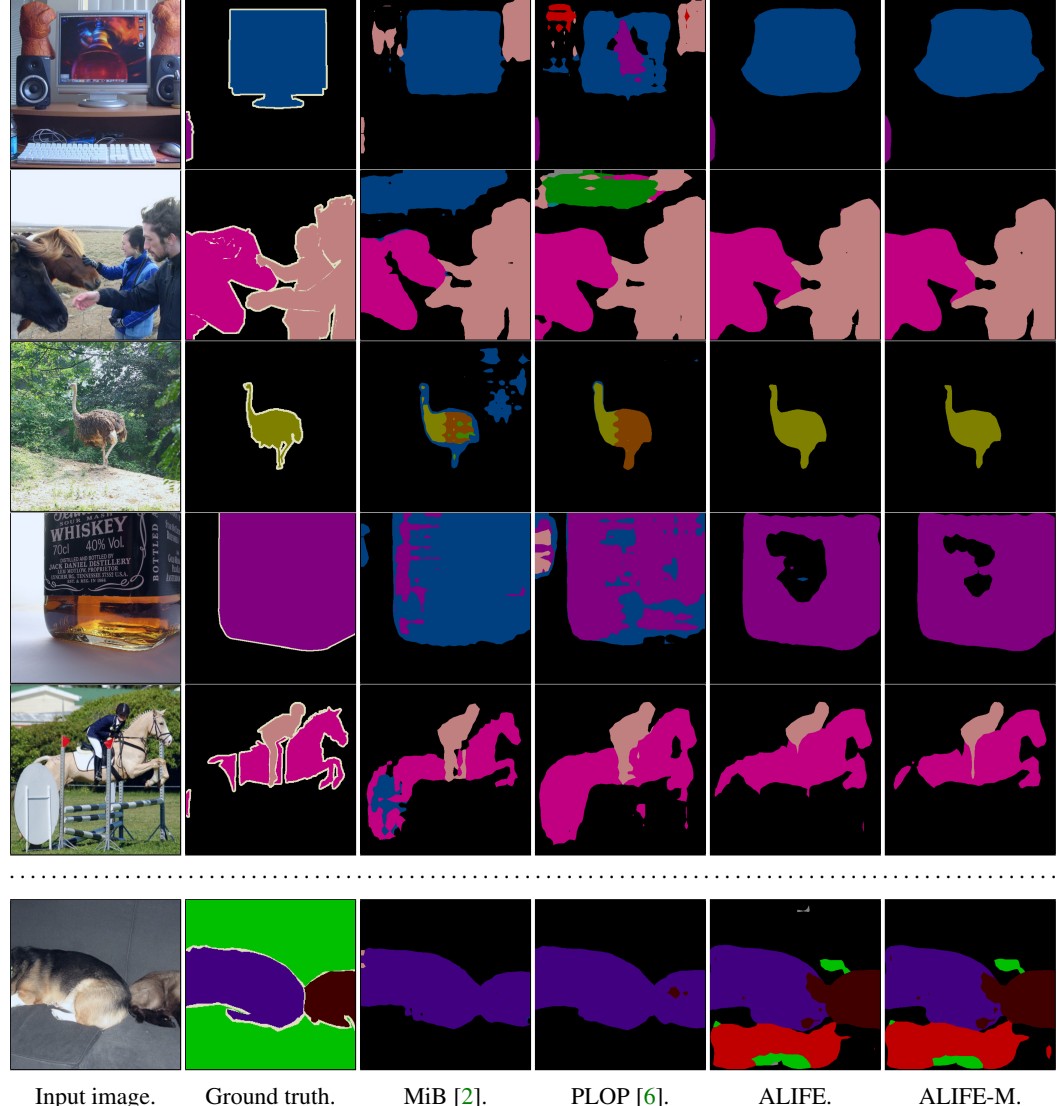

| Input image. | Ground truth. | MiB [2]. | PLOP [6]. | ALIFE. | ALIFE-M. |

Figure E: Visual comparison of ours and other methods [2, 6] on 20-1(1) for the PASCAL VOC [7] validation set. Each method is trained to recognize a tv category after learning 20 categories. The last row shows a failure case. Best viewed in color.

[5] Jia Deng, Wei Dong, Richard Socher, Li-Jia Li, Kai Li, and Li Fei-Fei. ImageNet: A large-scale hierarchical image database. In *CVPR*, 2009.

[6] Arthur Douillard, Yifu Chen, Arnaud Dapogny, and Matthieu Cord. PLOP: Learning without forgetting for continual semantic segmentation. In *CVPR*, 2021.

[7] Mark Everingham, Luc Van Gool, Christopher KI Williams, John Winn, and Andrew Zisserman. The PASCAL visual object classes (VOC) challenge. *IJCV*, 88(2):303–338, 2010.

[8] Kaiming He, Xiangyu Zhang, Shaoqing Ren, and Jian Sun. Deep residual learning for image recognition. In *CVPR*, 2016.

[9] Qibin Hou, Ming-Ming Cheng, Xiaowei Hu, Ali Borji, Zhuowen Tu, and Philip HS Torr. Deeply supervised salient object detection with short connections. In *CVPR*, 2017.

[10] Ahmet Iscen, Jeffrey Zhang, Svetlana Lazebnik, and Cordelia Schmid. Memory-efficient incremental learning through feature adaptation. In *ECCV*, 2020.

[11] Tsung-Yi Lin, Priya Goyal, Ross Girshick, Kaiming He, and Piotr Dollár. Focal loss for dense object detection. In *ICCV*, 2017.

[12] Andrea Maracani, Umberto Michieli, Marco Toldo, and Pietro Zanuttigh. RECALL: Replay-based continual learning in semantic segmentation. In *ICCV*, 2021.

[13] Umberto Michieli and Pietro Zanuttigh. Continual semantic segmentation via repulsion-attraction of sparse and disentangled latent representations. In *CVPR*, 2021.

[14] Adam Paszke, Sam Gross, Soumith Chintala, Gregory Chanan, Edward Yang, Zachary DeVito, Zeming Lin, Alban Desmaison, Luca Antiga, and Adam Lerer. Automatic differentiation in pytorch. In *NeurIPS Workshop*, 2017.

[15] Bolei Zhou, Hang Zhao, Xavier Puig, Sanja Fidler, Adela Barriuso, and Antonio Torralba. Scene parsing through ADE20K dataset. In *CVPR*, 2017.

**Algorithm 1** Pseudo code of incremental semantic segmentation with ALIFE.

1: // We omit a training process at a base stage ($t = 1$)
2: **for** $t \in \{2, \ldots, T\}$ **do**
3:  // Step 1
4:  $\{\phi^t, w^t\} \leftarrow \{\phi^{t-1}, w^{t-1}\}$ // Initialize a current model
5:  $\text{ep} \leftarrow 0$
6:  **repeat**
7:    Sample a mini-batch $\mathcal{B} \sim D^t$ // $\mathcal{B}$ indicates a mini-batch
8:    Update network weights of the current model $\{\phi^t, w^t\} \leftarrow \text{SGD}(\mathcal{B}, \{\phi^t, w^t\}, L_{\text{S1}})$ // $L_{\text{S1}}$ in Eq. (10)
9:    $\text{ep} \leftarrow \text{ep} + 1$
10:  **until** ep=EP // EP indicates the number of training epochs
11:
12:  // Step 2
13:  // Extract features which are used to replay in subsequent stages
14:  Freeze $\{\phi^t, w^t\}$
15:  **for** $c \in C_{\text{new}}^t$ **do**
16:    $T_c^t \leftarrow []$ // $T_c^t$ indicates a set of features for the category $c$
17:    $s \leftarrow 0$
18:    **repeat**
19:      $(x, y) \sim D^t$
20:      Extract a feature map $f^t \leftarrow \phi^t(x)$
21:      Average features for the category $c$ $m_c \leftarrow \frac{1}{|\mathcal{R}_c|} \sum_{\mathbf{p} \in \mathcal{R}_c} f^t(\mathbf{p})$
22:      $T_c^t \leftarrow [T_c^t; m_c]$
23:      $s \leftarrow s + 1$
24:    **until** $s = S$ // $S$ indicates the preset number
25:  **end for**
26:
27:  // Train rotation matrices
28:  Freeze $\{\phi^t, w^t\}$
29:  $\text{ep} \leftarrow 0$
30:  $\mathbf{R} \leftarrow \{\mathbf{R}_c \mid c \in C_{\text{prev}}^t\}$ // Initialize a set of rotation matrices randomly (See Eqs. (xi) and (xii))
31:  **repeat**
32:    Sample a mini-batch $\mathcal{B} \sim D^t$
33:    Update parameters of the rotation matrices $\mathbf{R} \leftarrow \text{SGD}(\mathcal{B}, \mathbf{R}, L_{\text{S2}})$ // $L_{\text{S2}}$ in Eq. (15)
34:    $\text{ep} \leftarrow \text{ep} + 1$
35:  **until** ep=10
36:
37:  // Step 3
38:  // Update saved features which are extracted in the previous stage
39:  Freeze $\mathbf{R}$
40:  **for** $c \in C_{\text{prev}}^t$ **do**
41:    // $M_c^{t-1}$ indicates a set of memorized features for the category $c$ in the previous stage
42:    Rotate features $\hat{M}_c^{t-1} \leftarrow \mathbf{R}_c M_c^{t-1}$
43:  **end for**
44:
45:  // Fine-tune a classifier
46:  Freeze $\phi^t$
47:  $\hat{\mathcal{M}}^{t-1} \leftarrow \{\hat{M}_c^{t-1} \mid c \in C_{\text{prev}}^t\}$
48:  $\text{ep} \leftarrow 0$
49:  **repeat**
50:    Sample a mini-batch $\mathcal{B} \sim D^t$
51:    Update classifier weights of the current model $w^t \leftarrow \text{SGD}(\{\mathcal{B}, \hat{\mathcal{M}}^{t-1}\}, w^t, L_{\text{S3}})$ // $L_{\text{S3}}$ in Eq. (19)
52:    $\text{ep} \leftarrow \text{ep} + 1$
53:  **until** ep=1
54:
55:  // Concatenate rotated features $\hat{M}_c^{t-1}$ and extracted features $T_c^t$
56:  **for** $c \in C_{\text{all}}^t$ **do**
57:    **if** $c \in C_{\text{prev}}^t$ **then**
58:      $M_c^t \leftarrow \hat{M}_c^{t-1}$
59:    **end if**
60:    **if** $c \in C_{\text{new}}^t$ **then**
61:      $M_c^t \leftarrow T_c^t$
62:    **end if**
63:  **end for**
64: **end for**