# OpenReview forum: "ALIFE: Adaptive Logit Regularizer and Feature Replay for Incremental Semantic Segmentation"
_NeurIPS.cc/2022/Conference — NeurIPS 2022 Accept_

### Official Review · Reviewer_hRHc · 2022-07-03

**Rating:** 6
**Confidence:** 5
**Soundness:** 2 fair
**Presentation:** 3 good
**Contribution:** 3 good

**Summary:**

The paper proposes a method for class-incremental learning in semantic segmentation. First, in a theoretical analysis of the gradients induced by commonly used loss functions are analyzed and several shortcomings are pointed out. As a way to mitigate these problem, a new knowledge distillation loss dubbed adaptive logit regularizer (ALI) and a feature replay strategy are proposed. Experimental evaluation shows the superiority of ALI over previous (non-replay-based) approaches in several experimental settings and competitive performance compared to other replay-based approaches, while having a much lower memory footprint.

**Questions:**

1. Eq. 9: I am not quite sure how this equation is derived from Table 2. Could the authors elaborate?
2. Have you varied the random seed between the three experiments that you average over?
3. Did you determine all method hyperparameters by cross-validation, in particular during the mentioned grid search mentioned in Appendix B?


**Limitations:**

Limitations have been discussed to some degree in the supplementary. However, I feel the hyperparameter sensitivity might be an additional limitation.

**Strengths And Weaknesses:**

Strengths:

1. Strong motivation of the method in the introduction and through the theoretical analysis of previously used loss functions.

2. Well-written related work section, which is informative in putting the paper into the larger context of incremental semantic segmentation. The novelty of the method is clearly stated and discussed w.r.t. other works.

3. Section 3.2 provides an overall very clear and informative mathematical description and analysis of de-facto standard loss functions in incremental semantic segmentation.

4. The experimental results are strong compared to previous methods and appear to be SOTA for non-replay-based methods.

Weaknesses:

1. CKD analysis: In the analysis of the CKD loss, it is not always clear to me, what the desired gradient behavior is vs. the observed behavior. Specifically, it would help, if on page 5, ll. 180-196, this would be clearly stated. Afterwards, it is also not completely clear to me how ALI solves these issues. This could be added in page 5, ll. 197-208 with reference to the desired behavior in the previously mentioned text part.

2. Loss application of ALI: It is not quite clear to me, why you use the vanilla CE and KD losses in Eq. 10 and also why you apply them specifically in these regions. I think this should be motivated in Section 3.2 and supported by an ablation study later on. In particular an ablation showing the experimental comparison with the commonly used CKD and CCE losses would help to better understand the benefits of ALI. Otherwise, also other factors such as better chosen hyperparameters could be the reason for the observed improvements through ALI.

3. Hyperparameter sensitivity. The method seems to contain a lot of hyperparameters, which even have to be determined for every incremental stage separately (cf. Table B in the appendix). I think this might limit the usability of the method, as it is computationally very expensive to run a grid search for each new incremental step. Maybe the authors could elaborate on this point?

4. Experimental setting: The approaches SSUL and RECALL compare on Pascal-VOC on the 10-1 setting and show particularly strong results in this setting. I think it would be better to evaluate this replay-based approach also in the most challenging settings introduced by other replay-based methods in incremental semantic segmentation.

5. Hyperparameter ablation studies: The influence of the different hyperparameters of the method is not shown. Table B of the appendix shows that different hyperparameters are chosen for each setting, so it would be good to know the influence of these hyperparameters on the final performance.

Overall, I really like the approach taken in this work. I also feel, the contributions contain a lot of value to the incremental and continual learning community. I would like to raise my rating, if my concerns are addressed satisfactory in the rebuttal.

Minor comments and suggestions:

6. Page 2, ll. 44-45: I think the CCE and CKD losses are indeed theoretically motivated. Rather than saying there is lack of theoretical support, I would rather rephrase to saying that the conducted theoretical analysis points out shortcomings which are addressed in this work.

7. Page 4, ll. 146: There might be a superscript t missing in the symbol R_new.

8. Page 4, ll. 162-166: Although it might appear trivial, it would be helpful to remind the reader that optimization is carried out in negative direction of the gradients shown in Table 1.

9. It would help a lot to state the dimensions of the features used for replay, mentioned on page 6, ll. 223-225 and in Eq. 12.

10. Eq. 12: The correlation score and the item index s have the same symbol, which is a bit confusing. Also, is the dot operator a scalar product between two feature vectors? Maybe this could be clarified.

11. I think the bold-face highlighting in Table 3 and 4 is a bit misleading. I would propose to compare replay-based methods and non-replay-based methods. In this sense, you would still have SOTA results for non-replay-based methods and competitive results among replay-based methods. But it seems unfair to compare Ours-S2 with, e.g., MiB or PLOP.

---

> ### Author Response · Authors · 2022-08-02
> **Response to Reviewer hRHc (3/3)**
>
> **[Q1]**- Eq. (9)
> - Since our goal is to find an objective function whose gradients are equivalent to the ones in Table 2, we have manually integrated the gradients to obtain such an objective (i.e., Eq. (9)). We will clarify this.
>
> **[Q2]**- Seed
> - We have generated a random seed for individual runs, and have reported the mean and standard deviation over three runs. We will clarify this.
>
> **[Q3]**- Did you determine all hyperparameters by cross-validation?
> - Thanks for your kind reminder. We have performed a grid search on the cross-validation set to find all hyperparameters, except \tau and \alpha in Eqs. (13) and (20), respectively. To be specific, the value of \tau is set to 10 in order to ensure that correlation scores are sharp enough. For the focal loss, we have used the default hyperparameter without tuning them (i.e., \alpha=2). We will clarify this.
>
> **[Minor comments]**
> - Thanks for the constructive comments. We will follow all your suggestions to improve the manuscript.
>
> - [L44-45 on page 2] We will revise the claims for lack of theoretical support.
>
> - [L146 on page 4] We will fix the typo $R_{new}$ into $R_{new}^{t}$.
>
> - [L162-166 on page 4] We will mention in Table 1 that the optimization process is carried out by stochastic gradient descent, i.e., SGD.
>
> - [The dimensions of features] Our approach memorizes $D$-dimensional convolutional features. We will mention the dimensions of features in L223-225 and Eq. (12) explicitly.
>
> - [Eq. (12) on page 6] We will differentiate the notation of the item index from the one of the correlation score.
>
> - [Tables 3 and 4] We will divide existing methods into two groups: one is non-replay-based methods and the other is replay-based ones.

---

> > ### Comment · Reviewer_hRHc · 2022-08-08
> > **Reply to Author Rebuttal**
> >
> > Thank you very much for the clarifications and for providing the additional experimental results regarding hyperparameters. I have no further questions and with the clarifications I do not have any major concerns anymore.

---

> ### Author Response · Authors · 2022-08-02
> **Response to Reviewer hRHc (2/3)**
>
> **[Hyperparameter sensitivity]**
> - [On ADE20K] For Step 1, we have performed a grid search on the cross-validation set to find values of $\lambda_{ALI}$ and $\lambda_{KD}$. To be specific, the search has been performed only once in the first incremental stage (i.e., $t=1$), since the hyperparameter search on ADE20K is computationally expensive. In the subsequent stages, we have used the same values of hyperparameters as shown in Table B of the supplementary material. For Step 2, we have performed a grid search for every incremental stage. This is because we fine-tune a classifier *only for a single epoch*, which is computationally acceptable. We will clarify this.
>
> - [On PASCAL VOC] For Step 1, we have additionally searched the number of training epochs along with two hyperparameters (i.e., $\lambda_{ALI}$ and $\lambda_{KD}$) for every incremental stage. To be specific, we have performed a grid search with 48 combinations of three hyperparameters, that is, {1,2,3} x {1,3,5,10} x {1,2,3,4} for $\lambda_{ALI}$ and $\lambda_{KD}$, and the number of training epochs, respectively. To traverse these combinations, we require 120 epochs (i.e., 120=12 + 12x2 + 12x3 + 12x4) in total. Although this sounds computationally expensive, we would like to note that the search cost of our approach is actually lower than MiB that tunes a single hyperparameter in the first incremental stage only. To find a value of $\lambda$ in Eq. (1) of the MiB paper, the authors define two hyperparameters A and B, where $\lambda = A \times 10^B$. They perform a grid search with 35 combinations of two hyperparameters, i.e., {1,2,3,4,5} x {-3,-2,-1,0,1,2,3} for A and B, respectively. Since MiB trains a model for 30 epochs, the search process requires 1,050 epochs in total. This requires 1.75 times more training epochs than our search cost even on 16-5(5) of PASCAL VOC. Note that we require 600 epochs in this case. For Step 2, we have performed a grid search to find the values of $\lambda_{ALI}$ and $\lambda_{KD}$. As aforementioned, the search cost for fine-tuning a classifier is relatively mild.
>
> - We also would like to note that the number of hyperparameters set by the grid search in our approach is comparable with other methods [9,30]. For example, PLOP has at least three hyperparameters (i.e., $S$ in Eq. (4), and two separate values of $\lambda$ in Eq. (9) for intermediate features and logits [9], while SDR contains at least four hyperparameters (i.e., $\lambda_{pm},  \lambda_{cl}$,  and $\lambda_{sp}$ in Eq. (1) and  $\lambda_{kd}$ in Eq. (2) [30]).
>
> **[Hyperparameter ablation studies]**
> - Following your suggestion, we analyze the robustness of our approach to different values of hyperparameters.
> |Ours-S1|100-50(1)|||||100-50(5)||||
> |:-:|:-:|:-:|:-:|:-:|-|:-:|:-:|:-:|:-:|
> |($\lambda_{ALI},\lambda_{KD}$)|mIoU$_{base}$|mIoU$_{new}$|mIoU|hIoU||mIoU$_{base}$|mIoU$_{new}$|mIoU|hIoU|
> |(1,1)|41.95|24.87|36.29|31.22||40.29|20.83|33.85|27.46|
> |(2,1)|42.16|23.60|36.01|30.25||40.87|21.18|34.35|27.90|
> |(3,1)|42.29|23.04|35.92|29.82||40.62|21.10|34.15|27.77|
> - From this table, we can see that our approach produces decent performance for different values of $\lambda_{ALI}$ on ADE20K.
>
> |Ours-S1|16-5(1)||||
> |:-:|:-:|:-:|:-:|:-:|
> |($\lambda_{ALI},\lambda_{KD}$)|mIoU$_{base}$|mIoU$_{new}$|mIoU|hIoU|
> |(1,1)|76.36|48.87|69.81|59.60|
> |(1,3)|77.05|49.23|70.43|60.08|
> |(1,5)|76.94|49.12|70.32|59.96|
> |(2,1)|77.53|51.73|71.39|62.06|
> |(2,3)|76.87|53.07|71.21|62.79|
> |(2,5)|76.41|51.21|71.06|61.60|
> |(3,1)|77.35|51.84|71.28|62.08|
> |(3,3)|77.86|52.46|71.81|62.68|
> |(3,5)|77.48|51.09|71.20|61.58|
> - Here all models are trained for 4 epochs. We can see from this table that our approach still shows the robustness to different values of$ \lambda_{ALI}$ and $\lambda_{KD}$ on PASCAL VOC.
>
> |Ours-S2|16-5(1)||||
> |:-:|:-:|:-:|:-:|:-:|
> |($\lambda_{ALI},\lambda_{MEM}$)|mIoU$_{base}$|mIoU$_{new}$|mIoU|hIoU|
> |(1,1)|77.42|53.46|71.72|63.25|
> |(1,10)|78.04|55.34|72.63|64.76|
> |(3,1)|77.69|54.01|72.05|63.72|
> |(3,10)|78.00|54.25|72.34|63.99|
> - This table shows that our approach to memorizing features is robust to changes of hyperparameters. We will add a discussion about this issue.

---

> ### Author Response · Authors · 2022-08-02
> **Response to Reviewer hRHc (1/3)**
>
> We sincerely appreciate the reviewer for insightful suggestions and constructive comments that help us to improve the manuscript. Below we address your questions one-by-one.
>
> **[CKD]**
> - The problem of CKD lies in how it transfers the knowledge from $p_{bg}^{t-1}$ into $p_{bg}^{t}$. In particular, CKD makes $p_{ckd}^{t}$ to imitate $p_{bg}^{t-1}$, instead of directly distilling from $p_{bg}^{t-1}$ to $p_{bg}^{t}$. This is because CKD assumes that new categories of the current stage $t$ are marked as the background at the previous stage $t-1$. Thus, the gradients in Eq. (8) should be removed, while the desired gradient should enable transferring the knowledge from $p_{bg}^{t-1}$ into $p_{bg}^{t}$ directly. We achieve this goal as in Table 2. To be specific, the gradient in the second row $p_c^{t}- p_c^{t-1}$ is now applied for all previous categories including the background one, enabling distilling from $p_{bg}^{t-1}$ to $p_{bg}^{t}$ directly. We will clarify this.
>
> **[ALI]**
> - The vanilla CE term requires ground-truth labels for training. Thus, it is available only for labeled regions, since unlabeled ones do not have annotations literally. Note that the CE term is essential for learning new categories, while ALI acts as a regularizer that prevents incremental semantic segmentation (ISS) models from overfitting to the new categories.
>
> - We would like to note that ALI always reduces logit values of new categories (See the first row of Table 2). Thus, if we apply ALI for all regions, the logit values of new categories decrease even for features within labeled regions. This hinders learning new categories severely, since labeled regions contain new categories only. Thus, we have applied ALI for unlabeled regions.
>
> - While distillation techniques [2,9,30] are typically applied for all regions, we would like to note that ALI already enables better transferring the knowledge for previous categories on unlabeled regions than the vanilla KD. Thus, additionally applying the vanilla KD for the unlabeled regions could be redundant. On the other hand, the KD term can be complementary to ALI for labeled regions. Specifically, using the KD term for labeled regions enforces a current model to produce probabilities similar to a previous one, even though features within those regions do not belong to previous categories. This allows us to further prevent the current model from being changed significantly, preserving the discriminative power for previous categories. We will clarify this.
>
> - We have shown in Fig.1(a) that Ours-S1 outperforms variants of CCE and CKD, and have also provided in Figure C of the supplementary material that Ours-S1 allows ISS models to obtain more balanced classifiers than CCE and CKD. In our response [Detailed comparison with CKD + CCE] to Reviewer PwVw, we provide more comparisons of Ours-S1 with CCE and CKD. Please refer to this.
>
> **[Experiment setting]**
> - Following the suggestion, we further present experimental results on the 10-1 setting (i.e., 11-10(10)).
> |Methods|mIoU$_{base}$|mIoU$_{new}$|mIoU|hIoU|
> |-|:-:|:-:|:-:|:-:|
> |MiB|12.25|13.09|12.65|12.66|
> |PLOP|**44.03**|15.51|30.45|22.94|
> |Ours-S1|37.49|**27.47**|**32.72**|**31.68**|
> |Ours-S2|43.75|34.28|39.24|38.43|
> |SSUL|71.31|45.98|59.25|55.91|
> |SSUL-M|**74.02**|53.23|**64.12**|**61.93**|
> |RECALL|65.00|**53.70**|60.70|58.81|
> - We can see that Ours-S1 outperforms MiB and PLOP in terms of mIoU and hIoU. In particular, Ours-S1 shows the remarkable hIoU gain (8.74%) over PLOP. The performance gains from Ours-S2 over Ours-S1 demonstrate the effectiveness of memorizing features once again. We can also see that SSUL and RECALL still outperform other methods including ours. However, as mentioned in L327-331 on page 9, we would like to note that our approach provides a better trade-off between accuracy and efficiency and is free to handle stuff categories. For example, RECALL is not applicable on ADE20K and SSUL-M shows worse results than Ours-S1 on ADE20K (See Table 3). In our response [Results on extremely challenging scenarios] to Reviewer eeB2, we also provide results of Ours-S1 on 100-50(10) of ADE20K. Please refer to this.

---

### Official Review · Reviewer_Rpor · 2022-07-10

**Rating:** 6
**Confidence:** 3
**Soundness:** 2 fair
**Presentation:** 2 fair
**Contribution:** 2 fair

**Summary:**

The catastrophic forgetting problem is particularly severe in ISS, since pixel-level ground-truth labels are available only for the novel categories at training time.
To address the problem, regularization-based methods exploit probability calibration techniques to learn semantic information from unlabeled pixels. While such techniques are effective, there is still lack of theoretical support. Replay-based methods propose to memorize a small set of images for previous categories. They achieve state-of-the-art performance at the cost of large memory footprint. The author propose in this paper a novel ISS method, dubbed ALIFE, that provides a better compromise between accuracy and efficiency. To this end, it first  show an in-depth analysis on the calibration techniques to better understand the effects on ISS. Based on this, it then introduce an adaptive logit regularizer (ALI) that enables our model to better learn new categories, while retaining knowledge for previous ones. It also present a feature replay scheme that memorizes features, instead of images directly, in order to reduce memory requirements significantly. Since a feature extractor is changed continually, memorized features should also be updated at every incremental stage. To handle this, it introduce category specific rotation matrices updating the features for each category separately. The paper demonstrate the effectiveness of our approach with extensive experiments on standard ISS benchmarks.

**Questions:**

questions are put in Strengths and Weakness.

**Ethics Review Area:**

["I don’t know"]

**Limitations:**

Yes

**Strengths And Weaknesses:**

Strengths of this paper are as follows:
1. The paper shows an in-depth analysis of probability calibration methods widely used for ISS, and introduces ALI that enables our model to better learn new categories, while maintaining the knowledge for previous categories.
2. The paper present a novel replay strategy using category-specific rotation matrices, which helps to alleviate catastrophic forgetting for previous categories with much less memory requirements than replaying raw images.
3. Extensive experiments demonstrate the effectiveness of our approach to using ALI and replaying  features with rotation matrices. It set a new state of the art on standard ISS benchmarks.

Weakness of this paper are as follows:
1. The proposed method  requires more memory than other similar methods [1, 4, 9] that do not adopt the experience replay.
2. The proposed method could not achieve the best performance on benchmark in Table 4, though the author claims that the proposed method need far less memory. But is it possible to increate the memory to achieve the best performance?
3. in line 137, the classifier weight is learnable, right? or it's fixed.
4. in line 147, how does the labeled regions R_new mean? is any region of image labelled and other regions not labelled? why there is labelled region and unlabelled regions? is it because the labelled regions are previous categories indicated region?
5. in line 264, in the fine-tuning step, is the rotation matrices fixed or still learnable?

---

> ### Author Response · Authors · 2022-08-02
> **Response to Reviewer Rpor**
>
> We sincerely appreciate the reviewer for the favorable consideration of our work. Below we address your questions one-by-one.
>
> **[More memory requirements]**
> - We have mentioned in the limitation part of the supplementary material (L174-180 on page 8) that our approach to memorizing features requires more memory requirements than non-replay-based methods [2,9,30]. However, we would like to emphasize that our method without memorizing features already outperforms the works of [2,9,30] on standard benchmarks by a large margin. Furthermore, since our approach achieves a better trade-off between accuracy and memory efficiency, we believe that it could be an effective and efficient alternative to the works of [3,26] that memorizes raw images.
>
> **[Q1]** - Is it possible to increase the memory to achieve the best performance?
> - A straightforward way to achieve the best performance in incremental semantic segmentation (ISS) is to replay previously seen images along with ground-truth labels during incremental stages. It however suffers from large memory requirements as well as data privacy issues. We would like to note that our feature replay scheme focuses on addressing these problems. Nonetheless, following the suggestion, we implement our method to exploit an experience replay as in SSUL-M that memorizes 100 previously seen images along with ground-truth labels, except that we do not use an off-the-shelf saliency detector. The results on 16-5(5) of PASCAL VOC are as follows:
> ||16-5(5)||||
> |-|:-:|:-:|:-:|:-:|
> |Methods|mIoU$_{base}$|mIoU$_{new}$|mIoU|hIoU|
> |Ours-S1-50|75.20|50.77|69.39|60.62|
> |Ours-S1-100|75.66|51.35|69.87|61.17|
> |SSUL-M|78.36|49.01|71.37|60.30|
> |RECALL|67.80|50.90|64.80|58.15|
> - Ours-S1-50 and -100 indicate that the number of memorized images is 50 and 100, respectively. To be specific, since memorized images can be directly used during Step 1, we have skipped Step 2 (i.e., fine-tuning classifiers). We can see that Ours-S1-50 already outperforms RECALL in terms of mIoU and hIoU. Note that RECALL uses 500 images for each previous category. We can also see that Ours-S1-50 and -100 show better hIoU scores than SSUL-M, although SSUL-M outperforms ours in terms of mIoU at the cost of using the off-the-shelf saliency detector.
>
> **[Q2]** - Is the classifier learnable in L137?
> - We have trained both a feature extractor and a classifier during Step 1 (See L140-142 on page 4). For memorizing features (L220-225 on page 6), we have fixed the feature extractor and the classifier. We have also fixed both the feature extractor and the classifier for training rotation matrices (See L263 on page 7). We will clarify this.
>
> **[Q3]** - A few questions in L147 on page 4
> - Thanks for your kind reminder. We will fix the typo $R_{new}$ into $R_{new}^{t}$. Note that $R_{new}^{t}$ indicates labeled regions as defined in Eq. (1). Please note also that training images in ISS are partially labeled to reduce the annotation cost. Specifically, pixels whose label belongs to current categories are marked only, while remaining ones are unlabeled. In the overlapped setting, the unlabeled regions could contain either previous and future categories (See L283-284 on page 8).
>
> **[Q4]** - Are the rotation matrices fixed in L264 on page 7?
> - To fine-tune the classifier, we have fixed the feature extractor and the rotation matrices. We will clarify this.

---

### Official Review · Reviewer_eeB2 · 2022-07-11

**Rating:** 6
**Confidence:** 4
**Soundness:** 3 good
**Presentation:** 3 good
**Contribution:** 3 good

**Summary:**

The paper addresses the question of incrementally adding new categories in a semantic segmentation problem. This is a variant of the more standard class incremental learning problem where data are annotated with the new categories only in each incremental session and may contain unannotated regions from seen or unseen categories.

The proposed approach makes two contributions: definition of a new term in the loss for regularization that takes into consideration the various types of data, and alignment of the memorized features from previous sessions to the new features using a category-specific rotation estimated from a criterion combining discriminating capacity and similarity with previous features. The approach is evaluated on two standard benchmarks in this field (PASCAL VOC and ADE20k).


**Questions:**

- It was not clear where the performances of competing approaches (SSUL, PLOP, etc.) come from. Did you re-implement the methods or take the number from the papers? Are the backbones identical?

- How can it be justified that the group of rotations, which are global isometries, is a good set of transformations ensuring feature alignment? How do the features behave when the number of incremental stages is large (I suspect that new features not only isometrically shift but may also be rearranged in a non-uniform way)? Maybe a T-SNE plot would give some idea about what is happening during sessions.


**Limitations:**

Non applicable

**Strengths And Weaknesses:**

Strengths

- Good and thorough discussion of loss terms in order to handle the various types of data and annotation state.
- Nice proposition to incrementally adapt memorized features using global alignment by rotations.
The ablation study, experiments and result analysis justify the approach on standard benchmarks.

Weaknesses
- Incremental work. Evolution of classical concepts of incremental learning: knowledge distillation and data replay.
- The maximal number of incremental stages is limited to 5: other papers (SSUL [3]) use more stages (until 11). Since the number of incremental steps is a critical parameter of incremental learning, in general, it would be fair to assess the approach on similar grounds.

---

> ### Author Response · Authors · 2022-08-02
> **Response to Reviewer eeB2 (2/2)**
>
> **[Q1-a]** - Numbers for other methods in Tables 3 and 4
> - In order to measure standard deviations over different seeds, we have implemented MiB and PLOP by using the official codes provided by the authors. The resultant values in Tables 3 and 4 are marked by $\dagger$. Numbers for other methods in Tables 3 and 4 are copied from SSUL. We will clarify this.
>
> **[Q1-b]** - Are the backbones identical?
> - Following the common practice, we have adopted DeepLab-V3 with ResNet101 pre-trained for ImageNet Classification. In particular, SDR and SSUL use ResNet101 provided by PyTorch, while MiB and PLOP exploit a variant of ResNet101 using in-place ABN [A]. Following SDR and SSUL, we have adopted ResNet101 provided by PyTorch. We will clarify this.
>
> - [A] In-Place Activated BatchNorm for Memory-Optimized Training of DNNs, CVPR 2018.
>
> **[Q2]** - How can it be justified that a group of rotations is a good set of transformations?
> - We agree in part with your point that features could change not only isometrically but also non-uniformly during incremental stages. This is because a feature extractor is updated continually. We however conjecture that the proposed ALI along with KD prevents the feature extractor from being updated significantly, since both ALI and KD encourage a current model to imitate a previous one. This could allow a global distribution of features for each category to be updated in a simpler (e.g., isometric) way. To verify this, we have shown in Table E of the supplementary material that FAN [19] in image classification shows inferior performance. Note that FAN updates memorized features *non-uniformly*, since it consists of multi-layer perceptrons with ReLU activations. As FAN does not differentiate categories, the superior performance of ours might come from the category-specific alignment. To further verify this, we exploit multiple FANs to update features for each category separately as in ours. Note that using multiple FANs for individual categories is computationally expensive compared to ours. The results on 16-5(1) of PASCAL VOC are as follows:
> |Methods|mIoU$_{base}$|mIoU$_{new}$|mIoU|hIoU|
> |-|:-:|:-:|:-:|:-:|
> |FAN|76.79|52.26|70.95|62.19|
> |Multiple FANs|77.83|52.98|71.91|63.04|
> |Ours-S2|78.04|55.34|72.63|64.76|
> - We can see that using multiple FANs show better results than FAN, suggesting that the category-specific alignment works favorably. We can also see that Ours-S2 still achieves the best performance, indicating that the isometric alignment is simple-yet-effective.
>
> - Although there is still a concern that the global distribution of features would also change non-uniformly as the number of incremental stages increases, our experiments have shown that mIoU and hIoU gains from memorizing features are rather remarkable on 16-5(5) of PASCAL VOC (See Table 4). Moreover, we have observed that Ours-S2 also achieves remarkable IoU gains even on the 11-10(10) case (Please see our response [Results on extremely challenging scenarios]). These empirical studies suggest that our approach to using rotation matrices ensures a good set of transformations for feature alignment. Nonetheless, we believe that a feature alignment scheme should update features globally as well as non-uniformly as your suggestion. We leave it for future work and will add a discussion on this issue.

---

> > ### Comment · Reviewer_eeB2 · 2022-08-09
> > **Comments on rebuttal**
> >
> > Thank you for providing detailed answers to my questions and supplementary experiments. Here are few comments on your replies.
> >
> > [Incremental work] Please don't be offended by this statement. By incremental I only meant that your work, which is done seriously, builds on existing concepts (improvement of distillation based learning loss and memory replay). I agree that this is a subjective statement, and that it depends on what one expects from contributions published in high ranked conferences such as Neurips.
> >
> > [Extremely challenging scenarios] I wouldn't call scenarios with 11 stages as "extremely challenging" but a standard setting of incremental learning. It seems that your approach is efficient for a small number of stages, but degrades more than several other methods when the number of stages increases (this was already noticeable in Table 4 for Pascal-VOC, and with a smaller amplitude for ADE20k on your new experiments). The reason for this behavior should have been analyzed better. You conjecture that "the proposed ALI along with KD prevents the feature extractor from being updated significantly, since both ALI and KD encourage a current model to imitate a previous one": I am not convinced this is the case using your rotation based memory alignment, and this is why I suggested to try to give some insight of the learned features dynamics while learning.

---

> ### Author Response · Authors · 2022-08-02
> **Response to Reviewer eeB2 (1/2)**
>
> We sincerely appreciate the reviewer’s time and effort to evaluate our work. Below we address your questions one-by-one.
>
> **[Incremental work]**
> - We respectfully disagree that our work is incremental. We would like to note that incremental semantic segmentation (ISS) methods [2,9,30] typically rely on distillation techniques to preserve the discriminative power for previous categories. In particular, they apply a probability calibration method to knowledge distillation as in Eq. (6). We have provided an in-depth analysis on the probability calibration method, and have pointed out that the calibration method has disadvantages in training ISS models. To address this, we have introduced a novel loss function, dubbed ALI, that enables the models to better learn new concepts, while preserving old ones. We do believe that our analysis together with ALI provides useful insights for other researchers who develop ISS methods. Also, although replay-based ISS methods [3,26] adopt a classical replay scheme, we argue that they have practical issues such as large memory requirements and data privacy. Our feature replay scheme is the first attempt to address these issues, achieving a better trade-off between accuracy and memory efficiency on standard benchmarks.
>
> **[Results on extremely challenging scenarios]**
> - Following the suggestion, we further provide results on 11-10(10) and 100-50(10) of PASCAL VOC and ADE20K, respectively. Our results are obtained for *a single run* due to the limited time during the rebuttal, while numbers for other methods are copied from SSUL.
> ||100-50(10)||||
> |-|:-:|:-:|:-:|:-:|
> |Methods|mIoU$_{base}$|mIoU$_{new}$|mIoU|hIoU|
> |PLOP|39.11|7.81|28.75|13.02|
> |SSUL|39.94|17.40|32.48|24.24|
> |Ours-S1|**40.01**|**17.62**|**32.60**|**24.47**|
> - We can see that Ours-S1 still outperforms SSUL in terms of all metrics. However, the gains over SSUL are not as drastic as in other cases on ADE20K, where Ours-S1 even outperforms SSUL-M (See Table 3). This implies that the 100-50(10) scenario is still quite challenging for existing methods including ours.
> ||11-10(10)||||
> |-|:-:|:-:|:-:|:-:|
> |Methods|mIoU$_{base}$|mIoU$_{new}$|mIoU|hIoU|
> |MiB|12.25|13.09|12.65|12.66|
> |PLOP|**44.03**|15.51|30.45|22.94|
> |Ours-S1|37.49|**27.47**|**32.72**|**31.68**|
> |Ours-S2|43.75|34.28|39.24|38.43|
> |SSUL|71.31|45.98|59.25|55.91|
> |SSUL-M|**74.02**|53.23|**64.12**|**61.93**|
> |RECALL|65.00|**53.70**|60.70|58.81|
> - We can see that Ours-S1 outperforms MiB and PLOP in terms of mIoU and hIoU. In particular, Ours-S1 shows the remarkable hIoU gain (8.74%) over PLOP. The performance gains from Ours-S2 over Ours-S1 demonstrate the effectiveness of memorizing features once again. We can also see that SSUL and RECALL still outperform other methods including ours. However, as mentioned in L327-331 on page 9, we would like to note that our approach provides a better trade-off between efficiency and accuracy and is able to handle stuff categories. For example, RECALL is not applicable on ADE20K and SSUL-M shows worse results than Ours-S1 on ADE20K (See Table 3).

---

### Official Review · Reviewer_PwVw · 2022-07-11

**Rating:** 8
**Confidence:** 4
**Soundness:** 4 excellent
**Presentation:** 4 excellent
**Contribution:** 4 excellent

**Summary:**

This paper describes an approach to Incremental Semantic Segmentation (ISS) of images. The authors provide a detailed and critical analysis of the Calibrated Cross Entropy (CCE) and Calibrated Knowledge Distillation (CKD) losses from the MiB approach [2], and from this analysis derive a new calibrated loss (called ALI) that combines the stability benefits of CCE with the forward transfer benefits of CKD. The authors additionally propose a three-stage incremental learning approach that first fits to a new task using ALI and (uncalibrated) CE and KD losses, then fits a rotation matrix to align memorized features with the new feature space, and finally fine-tunes the new task classifier on the current task samples and the newly rotated and memorized features from previous tasks. An extensive experimental evaluation is given on PASCAL VOC and ADE20K.


**Questions:**

+ **Improvement of S2 over S1**: on ADE20K the improvement of adding replay (and rotation and fine-tuning) is very marginal compared to that seen on PASCAL. This is also evident in the poor improvement (or even deterioration as noted in the paper) of SSUL-M with respect to SSUL. Do you have an explanation for this? Why would replay perform so poorly in some cases compared to others?

+ **mIoU scores over all**: I found the discussion of the drawbacks of the mIoU over *all* categories a bit hard to parse. Are the columns in Tables 3-4 the mIoU over all categories, as commonly reported in the literature? This should maybe be made more clear to facilitate comparison with existing work.


**Limitations:**

The authors include a thoughtful discussion of technical limitations in the Supplementary Material. I think something could and should be said about potential risks of incremental learning as well -- for example, a discussion of the dangers of biases becoming *baked in* to incrementally-learned representations, and the difficulty of eliminating them.

**Strengths And Weaknesses:**

# Strengths

+ **Clarity**: The paper is generally very well-written and the main contributions are clear and understandable. The paper was a pleasure to read and I am convinced it adds something interesting to the discussion on incremental learning.
+ **Motivation and technical development**: Similarly, the main technical motivations and developments are very clearly described, and the gradient analysis of CCE and CKD (as well as the proposed ALI) losses are quite compelling. The proposed approach is described in detail and I feel like the results presented would be easy to reproduce from the technical descriptions in the paper.
+ **Experimental evaluation**: The experimental evaluation is also quite convincing -- although see *Weaknesses* and questions below for some issues. The proposed approach demonstrates clear advantages over the state-of-the-art, both in terms of accuracy and memory efficiency.

# Weaknesses

+ **Steps**: Though the paper is well-written, there are quite a few moving parts and I feel that the generic use of "Step 1" and "Step 2" does some disservice to putting all of the pieces together into a coherent, global picture of the proposed approach. The approach seems more clearly articulated in **three** stages (names improvised): New Task Acquisition (currently Stage 1), Drift Compensation (via rotation), and maybe Current Task Fine-tuning.

+ **Detailed comparison with CKD + CCE**: This seems like a very important comparison as one of the main novelty claims is improvement over MiB which proposed these calibrated losses. The only direct comparison with these seems to be in Fig. 1, and thus gets kind of forgotten when arriving at the experimental evaluation. Moreover, the terminology is somewhat unclear: the proposed approach, I think, should be CE+KD+ALI (eq. 10).

+ **Ablations**: The feature alignment rotation is not ablated, moreover it would be interesting to see the ablations on ADE20K in attempt to understand why replay adds so little.

---

> ### Author Response · Authors · 2022-08-02
> **Response to Reviewer PwVw (2/2)**
>
> **[Q1]** - Why are the gains of memorizing images and features on ADE20K so insignificant?
> - It is obvious that ADE20K is more challenging than PASCAL VOC. In particular, ADE20K contains 35 stuff categories, while PASCAL VOC has a single background category. Since 1) the background category always belongs to base categories, and 2) most unlabeled pixels on PASCAL VOC belong to the background category during incremental stages, the unlabeled pixels are less likely to contain future categories even in the overlapped setting. By contrast, on ADE20K, the single background category is split into multiple stuff categories that could possibly belong to future categories. SSUL-M memorizes previously seen images, which contain unlabeled regions, together with ground-truth labels. Since pixels of stuff categories occupy about 60% of all the pixels on ADE20K (See Sec. 4 in [39]), the unlabeled regions could increase when the stuff categories belong to future ones. This might be problematic in that the number of labeled pixels decreases accordingly, lessening the effectiveness of the memoized images. In our case, the feature alignment scheme computes the correlation score between $f^{t-1}(\mathbf{p})$ and $m_c(s)$ as in Eq. (12). If the label of position $\mathbf{p}$ belongs to future categories, it might result in erroneous correlations, reducing the quality of feature alignment.
>
> **[Q2]** - Are the mIoU scores over all categories, denoted by mIoU in Tables 3 and 4, commonly used in the literature?
> - Thanks for your kind reminder. While all previous methods [2,3,9,25,28] report a mIoU score over all categories (denoted by mIoU in Tables 3 and 4), we have observed that the arithmetic mean over all categories does not reflect IoU scores of new categories well. Thus, we have proposed to report the harmonic mean, denoted by hIoU in Tables 3 and 4, as well as the mIoU score, which is commonly used in the literature. The motivation behind using the hIoU score is largely borrowed from the evaluation protocol of zero-shot learning methods (e.g., [A]). We will clarify this.
>
> - [A] Zero-Shot Learning-The Good, the Bad and the Ugly, CVPR 2017.
>
> **[Steps]**
> - Thanks for the valuable suggestion. We will revise Secs 3.2 and 3.3 following your suggestion.
>
> **[Potential risks]**
> - Thanks for the constructive suggestion. We will add a discussion on the potential issues in the limitation part.

---

> > ### Comment · Reviewer_PwVw · 2022-08-08
> > **Thank you for the clarifications and thoughtful conjectures.**
> >
> > The differences between ADE20K and PASCAL are indeed interesting, and I find the hypothesis put forward compelling. I encourage you to include some of this analysis in the main paper as it might point to interesting new directions on harder continual learning problems. The motivation for using hIoU (borrowed from ZSL) is also clear now.
> >
> > **To sum up**: The author rebuttal has addressed all questions and potential weaknesses that I identified in my original review, and I remain very positive about the quality and potential impact of this work.

---

> ### Author Response · Authors · 2022-08-02
> **Response to Reviewer PwVw (1/2)**
>
> We sincerely appreciate the reviewer for thoughtful and constructive comments that help us to improve the manuscript. Below we address your questions one-by-one.
>
> **[Detailed comparison with CKD + CCE]**
> - We will correct the typo “CE+ALI” into “CE+ALI+KD” in Fig. 1(a). We would like to note that the detailed comparison is also available in Fig. C of the supplementary material. This figure shows that “CE+ALI+KD” (denoted as “CE+ALI” due to the same typo) encourages classifier weights to be more balanced. To further compare our approach with variants of MiB, we provide the following results on 100-50(5) and 16-5(1) of ADE20K and PASCAL VOC, respectively. Numbers for MiB, which exploits CCE and CKD, are copied from the original paper.
> ||100-50(5)|||||16-5(1)||||
> |-|:-:|:-:|:-:|:-:|:-:|:-:|:-:|:-:|:-:|
> |Methods|mIoU$_{base}$|mIoU$_{new}$|mIoU|hIoU||mIoU$_{base}$|mIoU$_{new}$|mIoU|hIoU|
> |CE+KD|32.43|2.55|22.54|4.73||61.96|9.09|49.37|15.85|
> |CE+CKD|38.67|11.67|29.73|17.93||72.11|9.44|57.19|16.69|
> |CCE|10.19|4.57|8.33|6.31||63.25|38.53|57.37|47.89|
> |MiB(CCE+CKD)|38.21|11.12|29.24|17.23||75.50|49.40|69.00|59.72|
> |Ours-S1|**40.87**|**21.18**|**34.35**|**27.90**||**76.87**|**53.07**|**71.21**|**62.79**|
> - We can clearly see that training incremental semantic segmentation (ISS) models with CE+ALI+KD (denoted by Ours-S1) shows better results in terms of all metrics.
>
> **[Ablations]**
> - Following the suggestion, we conduct an ablation analysis on the objective function in Eq. (15). In particular, we show results on 100-50(5) and 16-5(1) of ADE20K and PASCAL VOC, respectively. Note that the results are obtained for *a single run* due to the limited time during the rebuttal.
> |||100-50(5)|||||16-5(1)||||
> |:-:|:-:|:-:|:-:|:-:|:-:|-|:-:|:-:|:-:|:-:|
> |$L_{FID}$|$L_{REG}$|mIoU$_{base}$|mIoU$_{new}$|mIoU|hIoU||mIoU$_{base}$|mIoU$_{new}$|mIoU|hIoU|
> |||40.92|21.03|34.33|27.78||76.61|53.10|71.02|62.72|
> ||o|41.04|21.23|34.48|27.98||76.75|54.17|71.38|63.51|
> |o||40.80|21.15|34.28|27.86||76.53|53.78|71.12|63.17|
> |o|o|**41.12**|**21.54**|**34.63**|**28.27**||**78.02**|**55.40**|**72.63**|**64.79**|
> - We can see from the first two rows that using the regularization term alone shows improvements in terms of all metrics. While the fidelity term in the third row produces better hIoU scores than the vanilla model (i.e., Ours-S1) in the first row, it shows slightly worse results in terms of mIoU$_{base}$ and mIoU. This could be due to the fact that maximizing the cosine similarity alone does not guarantee that the rotated prototypes are compatible with the current classifier (See L260-262 on page 7). The last row shows that both terms are complementary to each other. In our response [Q1], we describe the reason why the gains from memorizing features are lower on ADE20K than on PASCAL VOC.

---

> > ### Comment · Reviewer_PwVw · 2022-08-08
> > **Thank you for the new experimental results.**
> >
> > Thank you for the clarifications and the new experiments. I missed the analysis in Figure C, although I find this new table to be much more convincing. Again, the new ablations on ADE20K show very marginal differences when compared to PASCAL (see my response to the next comment).

---

### Meta-Review · Area_Chair_BUa1 · 2022-08-26

**Recommendation:** Accept
**Confidence:** Certain

**Metareview:**

This work deals with incremental semantic segmentation. The authors propose a three-step incremental learning approach. They provide an in-depth analysis of the probability calibration methods widely used for the ISS, and introduce an interesting proposal for incrementally adapting the memorized features using global alignment by rotations. They show strong results on standard benchmarks for incremental segmentation, including the ablation study.

The rebuttal provides valuable insight, and the questions raised by the reviewers have been convincingly answered by the authors.
On the whole, the reviewers converged positively, the novelty and the interest of the proposal stand out clearly.
Authors are encouraged to consider all comments for their final version.


**Award:**

No

---

### Decision · Program_Chairs · 2022-09-14

Accept